# InfoDLM: an Information-Adaptive Framework for Discrete Diffusion Language Model Pretraining

**Shirou Jing** [1]   **Chunshu Wu** [2]   **Chuan Liu** [1]   **Arghavan Bahadorinejad** [3]   **Feitong Qiao** [3]   **Dongfang Liu** [4]
**Tony Geng** [1]

## Abstract

Diffusion language models (DLMs) can match or surpass similarly sized autoregressive language models on language understanding and reasoning. However, their mask-and-denoise pretraining relies on heuristic random masking, which fails to target the most informative tokens. Consequently, the model spends significant computational effort on redundant or trivial tokens. To address this, we propose InfoDLM, an adaptive DLM pretraining framework that reformulates mask selection as an active, feedback-driven process. InfoDLM targets tokens that offer the highest measurable information gain during mask selection. Specifically, we: (1) introduce a Trainable Information-Gain (TIG) signal to quantify information gain of each masking configuration; (2) develop a feedback mechanism that adapts the masking policy to the model's evolving state with a maturity indicator; and (3) jointly optimize the DLM and masking policy through an interleaved training flow with only $\sim$14.1% wall-clock and $\sim$7.6% FLOPs overhead per cycle. Across reasoning-oriented benchmarks, InfoDLM achieves up to 13% improvement in reasoning accuracy over a small variant of LLaDA under comparable pretraining budgets.

## 1. Introduction

Language Models (LMs) are currently characterized by two primary paradigms: autoregressive LLMs (Brown et al., 2020; Kaplan et al., 2020; Hoffmann et al., 2022; Chowdhery et al., 2023) and the emerging field of Diffusion Language Models (DLMs) (Austin et al., 2021; Shi et al., 2024;

Lou et al., 2024; Ye et al., 2025). While autoregressive models dominated the field's early breakthroughs, DLMs are rapidly advancing as a competitive alternative. Recent developments, such as LLaDA (Nie et al., 2025a;b), demonstrate that discrete diffusion architectures can not only match but even surpass the performance of established autoregressive models across key benchmarks.

Intuitively, while autoregressive LLMs rely on sequential next-token prediction, DLMs mirror a fundamental cognitive process: reasoning. Rather than generating text in a single pass, DLMs "think" through a problem by iteratively polishing a complete sequence. This capability is driven by a mask-and-denoise pretraining objective, where the model reconstructs hidden segments of a sequence from the surrounding context.

Typically, the mask patterns and ratios are highly diverse in pretraining, presenting the model with a structured cognitive environment. By varying both the amount of hidden content and its specific placement, a broad distribution of completion tasks is induced, varying in difficulty and context configuration for each sequence (Prabhudesai et al., 2025). This diversity compels the model to develop a versatile reconstruction capability, as the model must learn to form a complete picture across varying levels of information scarcity and context configurations.

However, in existing efforts, the diverse mask selection is governed by heuristic random masking – while this approach covers a wide distribution of tasks, it emphasizes stochastic sampling rather than active focus. In human cognition, the development of reasoning is rarely a random act, but a selective process that prioritizes the most informative gaps in one's understanding. Based on this intuition, we hypothesize that DLM pretraining efficiency can be enhanced by a masking policy that selects highly-informative tokens.

To effectively prioritize information-rich tokens during DLM pretraining, three critical objectives must be addressed. (1) **Tractable Information Gain (IG) measurement**. Although IG exists as a theoretical foundation, its exact calculation is intractable. To enable feedback on token importance within a pretraining loop, it is essential to translate the

[1]Rice University, Houston, TX, USA [2]Pacific Northwest National Laboratory, Richland, WA, USA [3]Reinforce Labs, Palo Alto, CA, USA [4]Rochester Institute of Technology, Rochester, NY, USA. Correspondence to: Tony Geng <tg62@rice.edu>.

*Proceedings of the $43^{rd}$ International Conference on Machine Learning*, Seoul, South Korea. PMLR 306, 2026. Copyright 2026 by the author(s).

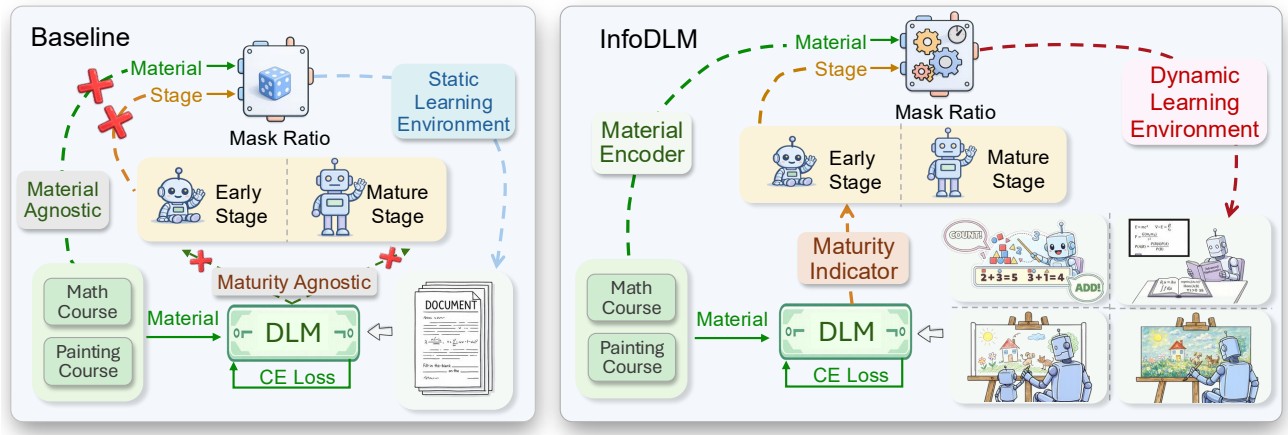

Figure 1. **Baseline vs. InfoDLM.** Baseline (left) construct learning environments with fixed or heuristic methods that are insensitive to the specific learning material and model maturity. InfoDLM (right) adapts mask choices to both the input material and maturity stage, generating more informative and dynamic learning environments.

theoretical principle into a computationally efficient form. (2) **Adaptive IG maximization**. Beyond measurement, an adaptive mechanism is required to dynamically adjust the masking policy according to the model's evolving state. This mechanism must strategically target tokens and contexts with the highest potential for information acquisition. (3) **Joint optimization of DLM and masking policy**. To achieve systemic convergence, a comprehensive training scheme is necessary to facilitate the stable co-evolution of the model and its masking policy. This framework must ensure that both components are optimized in a mutually reinforcing manner throughout the pretraining process.

To address the above challenges, we propose InfoDLM, a feedback-driven pretraining framework that transforms mask selection from a stochastic heuristic into an active, information-maximizing process tailored to enhance reasoning. As Figure 1 depicts, InfoDLM utilizes both material-intrinsic feedback and maturity indicators to modulate the masking ratio, creating a dynamically optimized learning environment for the DLM.

Compared to a baseline DLM, InfoDLM features three core technologies (detailed in Section 3). Specifically:

1. **Trainable Information-Gain (TIG):** A computationally efficient proxy for exact IG that quantifies the information acquisition of each masked example in real-time. TIG is lightweight enough to be integrated directly into the pretraining loop, allowing the masking policy to be trained via feedback without the prohibitive computational overhead of full-model backpropagation.

2. **Maturity-Aware Masking:** An adaptive mechanism that synthesizes TIG with training statistics into a low-cost model maturity indicator, dubbed "phase". Serving as a conditioning signal, phase allows the masking policy to dynamically maximize information acquisition by shift-

ing its focus from basic language structures to complex logical and numerical patterns as the model evolves.

3. **Interleaved Hybrid Training:** A joint optimization scheme that alternatively updates the DLM and the masking policy. Driven by real-time feedback signals, this scheme allows the DLM and policy to co-evolve, ensuring the model is consistently presented with challenging tasks while maintaining modest computational overhead.

Experimental results demonstrate that on 472M models, InfoDLM delivers up to 13% improvement in accuracy and faster convergence over LLaDA-472M across reasoning-oriented benchmarks.

## 2. Background

**LLaDA-style masking.** In this work, we leverage discrete diffusion language models (DLMs) in the LLaDA-style (Nie et al., 2025b) masking paradigm. Given a token sequence $x = (x_1, \ldots, x_T) \in \mathcal{V}^T$ for vocabulary $\mathcal{V}$, a forward step constructs a corrupted input by masking a subset of positions. Let $m \in \{0, 1\}^T$ be a token-wise mask with independent Bernoulli variables, where $m_i = 1$ indicates that $x_i$ is replaced by the special token [MASK] and $m_i = 0$ indicates that $x_i$ is revealed:

$$
\begin{aligned}
m_i &\sim \text{Bern}(\rho), \\
z &= (1 - m) \odot x + m \odot [\texttt{MASK}].
\end{aligned}
\tag{1}
$$

Here $\rho \in (0, 1)$ denotes the masking probability for position $i$, typically obtained by random sampling from $(0, 1)$, which is applied to each token. Then, a mask predictor $p_\theta(\cdot \mid z)$ with parameters $\theta$ is trained to recover the masked tokens conditioned on the corrupted input $z$. The standard objective computes cross-entropy only on masked positions:

$$
\mathcal{L}(\theta) = \mathbb{E}_{x,m}\left[-\sum_{i:\,m_i=0} \log p_\theta(x_i \mid z)\right].
\tag{2}
$$

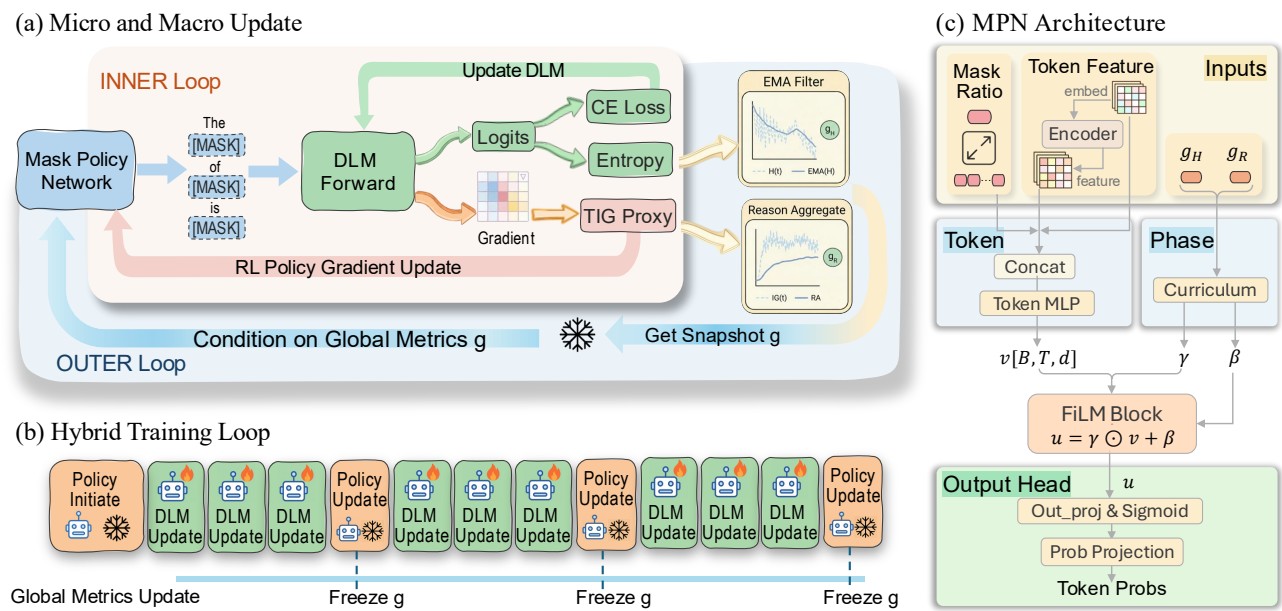

*Figure 2.* **Method overview. (a)** Micro–macro update scheme in our feedback-driven masking framework. The inner loop updates the DLM with cross-entropy and the mask policy with RL-style updates, while the outer loop tracks low-frequency global metrics $g$ for phase conditioning. **(b)** Interleaved hybrid training loop that alternates DLM and policy updates, keeping the other component fixed within each block. **(c)** Mask Policy Network (MPN) architecture. A token-wise branch outputs per-token mask probabilities, and a global branch embeds $g$ to generate FiLM parameters $(\gamma, \beta)$ while enforcing the target mask ratio.

**Information Gain (IG).** To quantify how informative an observation is for learning, a classical criterion is IG, widely used in Bayesian experimental design. Given prior knowledge $D$ and an additional observation $y$, IG is defined as the KL divergence from the posterior to the prior, equivalently the conditional mutual information between the model parameters and $y$ given $D$, measuring expected posterior uncertainty reduction:

$$\mathrm{IG}(y) = \mathrm{KL}(p(\theta \mid D, y) \,\|\, p(\theta \mid D)). \quad (3)$$

While this metric theoretically identifies the most informative observations, calculating the exact IG is intractable. This complexity prevents its direct application for real-time feedback during DLM pretraining. To overcome this, we derive a tractable, computationally efficient proxy for IG that provides real-time signals to guide a dynamic masking strategy, with minimal overhead incurred.

## 3. InfoDLM

**Overview.** Figure 2 summarizes the proposed feedback-driven DLM training framework. At each step, the policy samples a mask and the DLM runs a masked-prediction forward pass. From this forward pass, we compute (i) the masked-token cross-entropy to update the DLM and (ii) a lightweight TIG reward to optimize the policy. Meanwhile, we maintain low-frequency global statistics by exponentially smoothing predictive entropy and aggregating the TIG reward signal, producing a maturity metric $g = (g_H, g_R)$ that conditions the policy across training phases. We train the DLM and the policy with an interleaved schedule that

alternates CE updates and RL-style policy updates, holding the other component fixed within each block for training stability. We detail the step-wise TIG reward in Sec. 3.1, the maturity signal and phase conditioning in Sec. 3.2, and the interleaved hybrid training loop in Sec. 3.3.

### 3.1. TIG Reward for Mask Feedback

In this section, we present the development of the TIG reward through a top-down approach. We begin by directly defining TIG in its operational form. Then, we interpret IG in a reasoning relevant sense, providing the conceptual foundation for our metric. Finally, we provide the formal mathematical derivation that bridges the foundation with our lightweight TIG implementation.

To quantify how much information the model could acquire from each masked sample, we define a lightweight TIG reward that measures the informativeness of the designed conditional completion task for the current model.

For a batch $x$ and a sampled binary mask $m \in \{0,1\}^T$ under target mask ratio $\rho$, let $\mathcal{M} = \{i : m_i = 0\}$ denote the masked positions, we define TIG Reward as:

$$r_{\mathrm{step}}(x, m) = \sum_{i \in \mathcal{M}} \left\| p^{(i)} - e_{x_i} \right\|_2^2 \cdot \left\| h^{(i)} \right\|_2^2. \quad (4)$$

Here $h^{(i)}$ is the final-layer representation at position $i$, and $p^{(i)} = \mathrm{softmax}(W h^{(i)} + b)$ is the predictive distribution produced by the final layer in the same forward pass of $p_\theta(\cdot \mid z)$. We use $e_{x_i}$ as the one-hot target token, so for the per-token cross-entropy loss $\ell_i$, the readout-layer gradient satisfies $\|\nabla_W \ell_i\|_F^2 = \|p^{(i)} - e_{x_i}\|_2^2 \|h^{(i)}\|_2^2$.

We use this quantity as a low-cost proxy for per-token gradient energy to score each sampled mask. Intuitively, the score increases when the model incurs larger prediction error on masked tokens and the corresponding feature norms are larger, indicating higher potential learning gain from masking these positions. Therefore, we upweight such masks during policy optimization to steer training toward more informative completion tasks.

In the following paragraphs, we will establish the connection between IG and reasoning. A masked sample is considered beneficial if it improves a conservative proxy for conditional completion. We instantiate this proxy via proper scoring-rule regret. We then connect this regret reduction to information gain (IG) as an information-theoretic view criterion for selecting informative masks.

**Measuring reason by information gain.** Reasoning progress is not directly observable during pretraining, since what constitutes correct inference depends on latent structure that is not explicitly labeled. Therefore, we introduce a conservative proxy based on proper scoring-rule regret for the mask conditional completion task. Let $s(\cdot, y)$ be a strictly proper scoring rule, and denote the visible context by $c = x_{\setminus \mathcal{M}}$. We write $p^\star(\cdot \mid c) \triangleq p_c^\star$ for the (conceptual) conditional distribution of the masked tokens given the visible context. For the current model $p_\theta(x_{\mathcal{M}} \mid x_{\setminus \mathcal{M}}) = p_{\theta,c}$, we define a reasoning regret proxy as

$$\mathcal{R}_S(\theta) \triangleq \mathbb{E}_{(x,m)} \, \mathbb{E}_{y \sim p_c^\star} \Big[ s(p_c^\star, y) - s(p_{\theta,c}, y) \Big], \quad (5)$$

which is nonnegative and is minimized when $p_{\theta,c}$ matches the true distribution of masked tokens. Since mask predictions require non-local dependency modeling and evidence aggregation, improvements in this conditional completion task serve as a indicator of reasoning-relevant progress during pretraining, without relying on explicit supervision. Under local conditions (Appendix F, Eq. (37)), regret reduction is driven by posterior contraction of the model parameters, which can be quantified by information gain. Specifically, given current knowledge $D$ and an additional observation $y$, we use $\mathrm{IG}(y)$ as a proxy for scoring masks:

$$\mathrm{IG}(y) \triangleq \mathrm{KL}\big(p(\Theta \mid D, y) \, \| \, p(\Theta \mid D)\big) = I(\Theta; y \mid D). \quad (6)$$

To connect $\mathrm{IG}(y)$ to a tractable objective under masked prediction, we apply a Laplace approximation in Eq. 26 to the local $IG$, yielding a mutual-information surrogate whose dominant term depends on accumulated conditional Fisher information via a log-determinant form:

$$I(\Theta; D) \approx C + \frac{1}{2} \mathbb{E}_D \big[ \log \det \big( \Sigma_0^{-1} + \Delta(D) \big) \big], \quad (7)$$

$$\Delta(D) \triangleq \sum_{n=1}^{N} \mathcal{I}_{\mathrm{cond}}\big(\theta_0 \mid x_{\setminus \mathcal{M}}^{(n)}\big), \quad (8)$$

where $\mathcal{I}_{\mathrm{cond}}(\theta \mid x_{\setminus \mathcal{M}})$ denotes the Fisher information of the conditional likelihood $p_\theta(\cdot \mid x_{\setminus \mathcal{M}})$.

**Tractable TIG reward.** The log-determinant form in Eq. 7 makes the alignment to information gain explicit, but it is still too heavy to optimize at scale because it depends on a high-dimensional Fisher matrix and its spectrum. We therefore move to a scalar surrogate that preserves this alignment while being easy to estimate. Under a mild bounded-spectrum condition in Lemma F.1, the increase of the log-determinant can be related to a trace-based proxy (Eq. F), which motivates using $\mathrm{Tr}\,\mathcal{I}_{\mathrm{cond}}$ as our next step.

$$I(\Theta; y \mid D) \approx C' + \frac{1}{2} \mathbb{E}_D \left[ \frac{1}{\lambda} \mathrm{Tr}\big(\Delta(D)\big) \right],$$

$$\propto \mathbb{E}_y[\mathrm{Tr}\,\Delta(D)] = \sum_{n=1}^{N} \mathrm{Tr}\,\mathcal{I}_{\mathrm{cond}}\big(\theta_0 \mid x_{\setminus \mathcal{M}}^{(n)}\big). \quad (9)$$

Noting that for any vector $g$, $\mathrm{Tr}(gg^\top) = g^\top g = \|g\|_2^2$,

$$\mathrm{Tr}\,\mathcal{I} = \mathrm{Tr}\,\mathbb{E}_{x_{\mathcal{M}}} \big[ \nabla_\theta \log p_\theta \, \nabla_\theta \log p_\theta^\top \big]$$
$$= \mathbb{E}_{x_{\mathcal{M}}} \Big[ \big\| \nabla_\theta \log p_\theta(x_{\mathcal{M}} \mid x_{\setminus \mathcal{M}}) \big\|_2^2 \Big]. \quad (10)$$

Therefore, this yields an ideal gradient-energy reward $R_{\mathrm{ideal}}$ that is theoretically aligned with information gain:

$$R_{\mathrm{ideal}}(x, m) := \big\| \nabla_\theta \log p_\theta\big(x_{\mathcal{M}}^{\mathrm{obs}} \mid x_{\setminus \mathcal{M}}\big) \big\|_2^2. \quad (11)$$

However, computing $R_{\mathrm{ideal}}(x, m)$ at pretraining scale would require a full gradient with respect to *all* model parameters for every sampled mask, which is prohibitively expensive inside the INNER loop.

To obtain a tractable proxy, we exploit the structure of the final prediction layer: the readout-layer gradient energy factorizes as $\|\nabla_W \ell\|_F^2 = \|p - e_x\|_2^2 \|h\|_2^2$. Aggregating this per-token last-layer energy over $i \in \mathcal{M}$ yields our TIG Reward in Eq. 4. This proxy is computed from the same masked-denoising forward pass with no additional backpropagation for reward estimation.

Taken together, $r_{\mathrm{step}}$ is (i) *aligned* with information gain through gradient energy, and (ii) *cheap* enough to evaluate per batch, making it a practical feedback signal for selecting masks that maximize learnable progress at each step.

### 3.2. Maturity-Aware Policy Conditioning

With the TIG Reward in Sec. 3.1, the mask policy receives a principled feedback signal that prioritizes information-rich, mask-induced conditional completion tasks for the *current* DLM under a fixed masking budget. However, this signal entangles two sources of variation: fast per-batch fluctuations from input content, and slow drift from the DLM's evolving maturity. Without this separation, the policy cannot tell

whether a change in the signal reflects a harder example or a more competent model. We therefore condition the policy on an explicit low-frequency phase signal $g$, computed from inexpensive summary statistics of the DLM state, which acts as a dedicated global maturity indicator decoupled from token-level context and keeps the feedback loop stable and lightweight.

**Phase signal.** We define a low-frequency phase signal $g = (g_H, g_R)$ as a two-dimensional snapshot of the learner state, combining a globally smoothed entropy $g_H$ and a maturity signal that summarizes the reduction in reasoning regret. We derived the connection between $r_{step}$ and $\mathcal{R}_S$:

$$\mathcal{R}_S(t) \approx C \exp\left(-\beta \sum_{\tau=1}^{t} r_{step}(\tau)\right), \qquad (12)$$

here $t$ denotes training steps (See more in Appendix G). Concretely, let $H_{step}$ denotes the batch-level predictive entropy computed from the current DLM outputs on the masked denoising task, and let $r_{step}$ be the per-batch TIG Reward, we form the phase signal $z = (g_H, g_R)$ as:

$$g_H \leftarrow \text{EMA}(H_{step}), \quad g_R \leftarrow -\beta \log \sum_{k=1}^{K} r_{step}^{(k)}, \quad (13)$$

where $\text{EMA}(\cdot)$ is an exponential moving average filter, and the aggregation in $g_R$ is taken over the most recent $K$ training batches to obtain a stable progress indicator.

Intuitively, the entropy term $g_H$ monitors the model's uncertainty under the current masking distribution, while the reward term $g_R$ tracks how informative the masked instances have been for contracting regret. We separate the lifecycle of $g$ into two distinct operations to keep the conditioning low-frequency. During DLM updates, we maintain a running accumulator of $(H_{step}, r_{step})$ that refreshes its internal statistics every $K=5$ steps; this accumulator is *not* read by the policy and the snapshot $g$ visible to the policy is held fixed throughout the DLM block. At each DLM→policy transition we publish the accumulated statistics into a new snapshot $g = (g_H, g_R)$, which is then held constant for the entire subsequent policy-update block. This freeze-on-transition design ensures the policy never sees $g$ drifting within its own update window, which would otherwise inject the same kind of nonstationarity into the policy gradient that $r_{step}$ already carries through the DLM. Conditioning on $g$ allows the policy to represent phase-dependent masking directly, rather than relying on a static masking rule applied across all phases.

**Policy architecture with FiLM conditioning.** To incorporate the phase signal, we implement a MLP-based simple yet effective mask policy $\pi_\psi(m \mid emb(x), g, \rho)$ with three input branches: token features $emb(x)$, phase signal $g = (g_H, g_R)$, and target mask ratio $\rho$. The token branch

forms a thorough token representation by projecting the token embedding and a contextual token encoding into a shared feature space; in parallel, $\rho$ is broadcast to the same feature width and concatenated as a budget cue. To let the mask patterns adjust across maturity stages change, without acting as another token-level feature that can dominate local mask decisions, the phase branch maps $g$ to FiLM parameters that modulate the token representation,

$$v = \text{MLP}\left(\left[\text{Proj}_e(\text{Emb}(x)) \; ; \; \rho \, \mathbf{1}_T\right]\right), \qquad (14)$$

$$Logits = \text{out}\left[\gamma(g) \odot v + \beta(g)\right], \qquad (15)$$

where $v$ is the token level latent representation to modulate.

We stabilize this modulation by initializing $\gamma$ near identity and using a positive-bounded nonlinearity, so training starts effectively unconditioned and phase effects appear only once $v$ is reliable. Finally, we enforce the budget through a per-sample bias $\lambda$ found by bisection so that $\text{sigmoid}(\text{Logits} + \lambda)$ has mean equal to $\rho$ over the sequence; the discrete mask $m$ is then drawn by independent Bernoulli sampling from these projected probabilities, retaining token-level flexibility while matching the target ratio. Overall, this design keeps conditioning low-cost and stable, enables phase-adaptive masking under a fixed budget, and mitigates early instability from overly strong conditioning.

### 3.3. Hybrid Interleaved Training Flow

The mask policy controls the distribution of masks, which in turn defines the conditional completion tasks used to train the DLM. Optimizing the policy and the DLM simultaneously will create a strongly nonstationary feedback loop: Policy updates change the training environment, while DLM updates change the reward landscape and the phase signal computed from the same environment. As a result, the TIG Reward and the phase snapshot can drift within the same update window, making policy gradients noisy and training unstable. We therefore use an interleaved freeze-and-update schedule that alternates between updating the DLM with a fixed policy and updating the policy with a fixed DLM.

Concretely, training proceeds in alternating phases. During a DLM-update phase, we freeze the mask policy, sample masks under the target ratio, and update the DLM parameters by the standard masked-token cross-entropy objective on the mask prediction tasks. During a policy-update phase, the mask $m$ is a high-dimensional discrete sample from $\pi_\psi(\cdot)$, so the TIG objective is an expectation over stochastic actions and does not yield a direct differentiable loss in $\psi$. We therefore freeze the DLM, compute the TIG Reward from the same forward pass, and update the policy with REINFORCE. To ensure the conditioning is low-frequency and stable, we maintain the phase signal online only during DLM training and pass the fixed snapshot $g$ to the policy throughout each policy-update phase.

We formalize the DLM update objective under the masks sampled from the frozen policy. Given $z = (1 - m) \odot x + m \odot [\texttt{MASK}]$, $m \sim \pi_\psi(\cdot \mid x, z, \rho)$, real mask probability for each token $p_i$, the DLM is trained by the cross-entropy loss:

$$\mathcal{L}_{\text{DLM}}(\theta) = \mathbb{E}_{x \sim \mathcal{D}, m} \left[ -\frac{1}{T} \sum_{i=1}^{T} \frac{1}{p_i} \log p_\theta(x_i \mid z) \right]. \tag{16}$$

Policy optimization is non-differentiable through the discrete mask samples, so we update $\psi$ with REINFORCE using the TIG Reward $r_{\text{step}}(x, m)$. With a variance-reduction baseline $b$, the policy gradient is form as:

$$\nabla_\psi \mathbb{E}[r_{\text{step}}(x, m)] = \mathbb{E}\Big[\big(r_{\text{step}}(x, m) - b\big) \\ \nabla_\psi \log \pi_\psi(m \mid x, z, \rho)\Big]. \tag{17}$$

To prevent early collapse, we first regularize the policy toward the target ratio with a soft budget loss, and then apply the hard ratio-preserving projection $\pi_\rho$ for sampling. This is because in early training, $r_{\text{step}}$ tends to favor extreme masking, which would otherwise make $\pi_\rho$ degenerate into a near random mask distribution. The policy is therefore trained with three terms that we now define explicitly. The information-gain surrogate $\mathcal{L}_{IG}$ is the REINFORCE objective from Eq. 17,

$$\mathcal{L}_{IG}(\psi) = - \mathbb{E}_{x,m}[(r_{\text{step}}(x, m) - b) \log \pi_\psi(m \mid x, z, \rho)]. \tag{18}$$

The entropy regularizer $\mathcal{L}_{ENT}$ acts on the per-token Bernoulli logits of the policy itself and discourages early deterministic collapse,

$$\mathcal{L}_{ENT}(\psi) = -\lambda_H \, \mathbb{E}_{x,z} \left[ \frac{1}{T} \sum_{i=1}^{T} \mathcal{H}(\text{Bern}(p_{\psi,i})) \right], \tag{19}$$

where $p_{\psi,i}$ is the policy's per-token mask probability. The soft budget loss $\mathcal{L}_{Budget}$ pulls the expected mask rate toward the target ratio $\rho$,

$$\mathcal{L}_{Budget}(\psi) = \lambda_B \left( \mathbb{E}_{x,z} \left[ \frac{1}{T} \sum_i p_{\psi,i} \right] - \rho \right)^2. \tag{20}$$

The full policy loss is the sum of the three terms,

$$\mathcal{L}_{Total} = \mathcal{L}_{IG} + \mathcal{L}_{ENT} + \mathcal{L}_{Budget}. \tag{21}$$

## 4. Experiments

### 4.1. Setup

**Data and benchmarks.** We pretrain all models on SlimPajama-627B, a large-scale filtered web corpus, using a shared tokenizer and data mixture. We then evaluate INFODLM on a suite of reasoning benchmarks in both direct inference and supervised fine-tuning settings, covering math reasoning problems(GSM8K) and language understanding(ANLI, SNLI) used in the SFT stage. A detailed description of all datasets is provided in Appendix B.

*Table 1.* Per-step cost of INFODLM vs. the SMDM baseline at 472M scale. The baseline has no MPN phase. The bottom row aggregates one 5:1 DLM/MPN cycle.

| | Baseline | INFODLM |
|---|---|---|
| DLM Time/step (s) | 2.02 | 2.09 (+3.5%) |
| DLM FLOPs/step ($\times 10^{14}$) | 9.32 | 9.99 |
| MPN Time/step (s) | — | 1.07 |
| MPN FLOPs/step ($\times 10^{14}$) | — | 0.186 |
| Per cycle Time (s) | 10.10 | 11.52 (+14.1%) |
| Per cycle FLOPs ($\times 10^{14}$) | 46.6 | 50.14 (+7.6%) |

**Implementation details.** We build INFODLM on the SMDM (Nie et al., 2025a) codebase and instantiate models at 472M and 1B parameters, with an MPN of 1.2M parameters. INFODLM-1B is trained for 220k steps with AdamW using learning rate $2 \times 10^{-4}$, global batch size 576, and sequence length 2048, totaling roughly 220B pretraining tokens. The 472M model is trained for 60k steps over $\sim$35B tokens with global batch size 288 at the same sequence length. We adopt an interleaved schedule with an initial MPN warm-up followed by alternating DLM and MPN update blocks. For supervised fine-tuning, we use a global batch size of 896 and sequence length 256. Full optimization and compute details are provided in Appendix B.

### 4.2. Main Results on Reasoning Benchmarks

To assess reasoning performance, we evaluate INFODLM against SMDM (Nie et al., 2025a), a smaller variant of the state-of-the-art discrete DLM LLADA (Nie et al., 2025b) under matched pretraining token budgets. Table 2 reports both SFT-evaluated reasoning benchmarks and zero-shot tasks (ARC-e, ARC-c, HellaSwag, WinoGrande) at 472M and 1B scales. At 472M, INFODLM delivers consistent gains across all SFT benchmarks and zero-shot reasoning tasks. At 1B, the same trend holds for SFT and is reinforced in zero-shot: ARC-c gain $+2.8\,$pp and HellaSwag gains $+5.9\,$pp. Overall, the gains are pronounced on reasoning-heavy benchmarks, supporting the claim that the learned masking policy reallocates pretraining signal toward infor-

*Table 2.* Downstream evaluation across 472M and 1B scales. SFT covers GSM8K, ANLI, SNLI; zero-shot covers ARC-e, ARC-c, HellaSwag, WinoGrande.

| Benchmark | SMDM | InfoDLM | SMDM | InfoDLM |
|---|---|---|---|---|
| | **SFT-472M** | | **SFT-1B** | |
| GSM8K | 40.48 | **52.03** | 57.09 | **57.24** |
| ANLI | 35.50 | **49.30** | 43.40 | **49.50** |
| SNLI | 85.65 | **87.79** | 88.70 | **89.80** |
| | **Zero-shot-472M** | | **Zero-shot-1B** | |
| ARC-e | 31.65 | **35.47** | 38.68 | **38.85** |
| ARC-c | 19.79 | **23.37** | 24.31 | **27.13** |
| HellaSwag | **30.04** | 27.99 | 36.48 | **42.37** |
| WinoGrande | **50.67** | 50.04 | 49.50 | **50.59** |

mative, reasoning-relevant tokens.

## 4.3. Mask Entity Analysis

To illustrate how the learned policy evolves during training, we analyze mask selections from INFODLM-472M using Universal Dependencies (UD) part-of-speech tags (De Marneffe et al., 2021). For a fixed ANLI example, early-stage masks are relatively diffuse, whereas later they concentrate on key reasoning tokens while largely avoiding function words and punctuation, as shown in Figure 3(a). A similar trend appears in Figure 3(b): as the mask policy and the DLM co-evolve, mask probability increases for reasoning-relevant categories such as adverbs (ADV), verbs (VERB), and subordinating conjunctions (SCONJ), and decreases for low-information categories such as spaces (SPACE) and punctuation (PUNCT). Further quantitative analysis of context-adaptivity and stage-adaptivity of the learned policy, together with full UPOS statistics, is provided in Appendix D.

*Figure 3.* **Mask policy analysis.** (a) Token-level masking scores from the TIG reward (orange = high, blue = low). (b) Mask ratios on selected UPOS tags; INFODLM increases masking on reasoning-related tokens (ADV/VERB/SCONJ) and decreases it on SPACE/PUNCT compared to random masking.

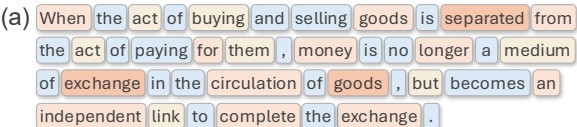

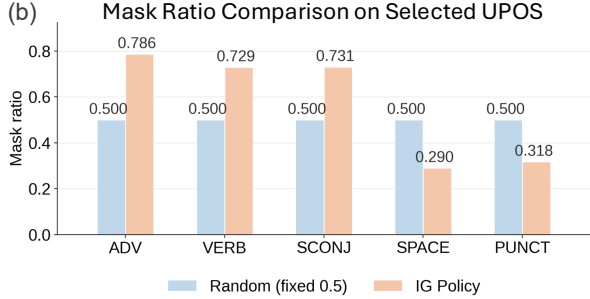

## 4.4. Mask Policy Analysis and Ablations

We view the mask as a learning environment for the DLM. To characterize its effect, we analyze INFODLM's masking policy from two angles, comparing different masking strategies and ablating key components of INFODLM.

**Masking strategy ablation.** To demonstrate that different masking policies can materially influence how a DLM learns, we isolate the effect of the masking *strategy* by varying only the policy used to sample masks. We train the InfoDLM-472M under five masking strategies: (i) RANDOM sampling (Nie et al., 2025a); (ii) SPAN masking with span length of 2∼5 (Joshi et al., 2020); (iii) FREQUENCY-informed masking (Kosmopoulou et al., 2025) which uses corpus token-frequency statistics to selectively mask tokens;

(iv) PMI-informed masking (Levine et al., 2021), which uses the Point-wise Mutual Information method; and (v) INFODLM(ours). We report accuracy on two representative downstream tasks (GSM8K and ANLI) under different pretraining token budgets.

Table 3 shows that SPAN masking yields light gains over purely random masking, while heuristic, statistics-informed strategies (FREQUENCY, PMI) further improve downstream reasoning performance under the same token budget. In contrast, INFODLM consistently attains the best GSM8K and ANLI performance. It also reaches a comparable accuracy level with fewer pretraining tokens, e.g., INFODLM trained on 10B tokens matches or exceeds baselines trained on 35B tokens. This pattern supports our view that the masking strategy shapes the learning environment. SPAN changes context structure but model-agnostic, while FREQUENCY/PMI leverage corpus statistics to better target informative gaps. INFODLM goes one step further by learning from feedback which inference gaps are most beneficial for the current model, which may help explain its advantage under constrained pretraining budgets.

**Method ablation.** We validate the contribution of each design component by incrementally adding our proposed components to SMDM, resulting in three configurations (A–C). Two further ablations on the RL optimizer choice and training-schedule sensitivity are deferred to Appendix E. In setting (A), we replace random masking with a policy-gradient–trained masking policy optimized using the TIG Reward defined in Sec. 3.1. As shown in Table 4, introducing the TIG Reward yields a substantial improvement and enables the model to reach a target accuracy with fewer pretraining tokens. In setting (B), we additionally adopt an cyclic training schedule that alternates between updating the policy and the DLM, while omitting phase conditioning. This dynamic schedule yields little to no improvement, suggesting that directly alternating updates can make it harder for the masking policy to track the DLM's rapid progress and adjust accordingly, leading to performance that is comparable to, or slightly worse than, the non-alternating setting. Finally, in setting (C), we feed the aggregated phase signal from Sec. 3.2 into the mask policy network via FiLM-style conditioning, yielding the full INFODLM configuration. As shown in Table 4, adding this maturity-aware phase conditioning further improves benchmark performance, suggesting that phase-aware control helps the MPN adjust masking choices as the DLM evolves and helps stabilize the co-evolution between environment construction and model learning under both limited and full token budgets.

**Scaling with pretraining tokens.** To illustrate how different masking strategies affect scaling with the pretraining token budget, we plot validation-loss curves in Figure 3(b) and Figure 4(b). We report validation loss here as a *curriculum-dynamics diagnostic* rather than a measure of

*Table 3.* Method ablation on the 472M model. Each cell denotes accuracy ↑. Left: GSM8K and ANLI accuracy of different masking policy under different pre-training token budgets. Right: validation loss vs. training TFLOPs for the same variants.

| # training tokens | GSM8K | | ANLI | | |
|---|---|---|---|---|---|
| | 20B | 35B | 10B | 20B | 35B |
| Baseline (SMDM) | 40.29 | 40.48 | 33.30 | 35.10 | 35.50 |
| Span masking | 41.88 | 41.42 | 35.90 | _36.00_ | 36.50 |
| Frequency masking | _44.14_ | 42.27 | 38.30 | 34.70 | 36.50 |
| PMI masking | 40.25 | _42.54_ | _39.00_ | 35.40 | _39.60_ |
| InfoDLM (ours) | **51.78** | **52.04** | **41.20** | **47.60** | **49.30** |

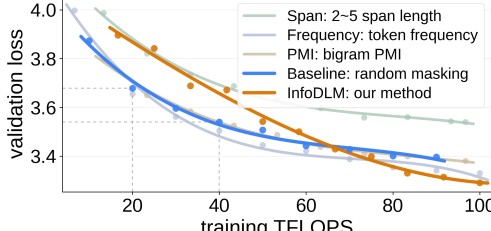

*The best and second results are highlighted in **boldface** and underlined.*

*Table 4.* Method ablation on the 472M model. Each cell denotes accuracy ↑. Left: GSM8K and ANLI accuracy under different pre-training token budgets as we add three step ablations. Right: validation loss vs. training TFLOPs for the same variants.

| # training tokens | GSM8K | | ANLI | | |
|---|---|---|---|---|---|
| | 20B | 35B | 10B | 20B | 35B |
| (O) Baseline (SMDM) | 40.29 | 40.48 | 33.30 | 35.10 | 35.50 |
| (A) + TIG reward | _50.8_ | _51.07_ | _45.80_ | 40.50 | _47.20_ |
| (B)    + Cyclic training | 48.52 | 50.61 | **46.10** | _42.40_ | 46.70 |
| (C)      + Phase signal (full InfoDLM) | **51.78** | **52.04** | 41.20 | **47.60** | **49.30** |

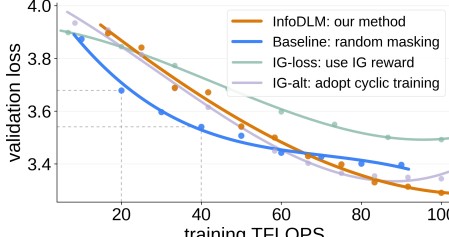

*The best and second results are highlighted in **boldface** and underlined respectively.*

final quality (covered by the downstream metrics in Table 2): the characteristic start-high, end-low crossover predicted by curriculum learning is directly visible in the loss trajectory but is washed out by saturation in downstream metrics. For evaluation, we use matched masks across methods when computing validation loss. We observe that INFODLM shows a slower early decline in loss but continues to improve later in training, whereas other strategies gradually plateau. This suggests that our masking policy adapts to the evolving learner and reallocates a fixed masking budget toward informative inference gaps, rather than masking already predictable tokens that yield diminishing training signal.

**TIG reward trend.** We plot the TIG Reward (EMA) against training compute (TFLOPs) in Figure 4. INFODLM achieves higher rewards early and continues to grow throughout pre-training, whereas other masking strategies plateau. Since $r_{step}$ weights prediction error $\|p^{(i)} - e_{x_i}\|_2^2$ by feature magnitude $\|h^{(i)}\|_2^2$, indicating that INFODLM continues to place currently mispredicted and represented with high leverage in the last layer. Coupled with phase conditioning, this suggests the MPN tracks shifts in where gradient energy concentrates as the DLM matures, which is consistent with the larger downstream reasoning gains we observe, especially under tighter token budgets.

**TIG proxy fidelity.** We systematically compare TIG, which uses only head-layer gradient energy, against a higher-fidelity IG estimate computed from the full 18-layer Fisher information with 64 candidate masks per sample, across training checkpoints in Table 5. TIG–HFIG correlation starts at $\rho = 0.592$ and rapidly improves to $\rho = 0.756$ by cycle 4, remaining strong ($\rho \geq 0.72$) throughout training. Top-30 overlap exceeds 0.82 from cycle 2 onward, confirming

that TIG captures the coarse difficulty structure needed to direct the masking policy without requiring exact per-token rankings. Table 6 reports Spearman $\rho$ across a parameter-subset fidelity ladder; fidelity increases monotonically with included layers, while the head-only proxy still achieves $\rho = 0.59$–0.76 across training, offering a favorable fidelity–cost tradeoff.

*Table 5.* TIG proxy fidelity: rank correlation with full-model IG across checkpoints. Best per column in **bold**.

| Stage | Spearman | Kendall | Top-5 | Top-10 | Top-30 |
|---|---|---|---|---|---|
| cycle 0 | 0.592 | 0.479 | 0.378 | 0.556 | 0.751 |
| cycle 4 | **0.756** | **0.632** | **0.515** | **0.726** | 0.853 |
| cycle 8 | 0.730 | 0.602 | 0.505 | 0.707 | 0.821 |
| cycle 11 | 0.721 | 0.599 | 0.560 | 0.687 | **0.854** |

*Table 6.* Parameter-subset fidelity ladder (Spearman $\rho$). Rows add progressively more transformer layers to the TIG proxy.

| Layers | cycle 0 | cycle 4 | cycle 8 | cycle 11 |
|---|---|---|---|---|
| Head-only | 0.592 | 0.756 | 0.730 | 0.721 |
| +last 3 layers | 0.684 | 0.808 | 0.777 | 0.758 |
| +last 6 layers | 0.819 | 0.880 | 0.855 | 0.823 |
| +last 9 layers | 0.924 | 0.914 | 0.893 | 0.878 |

# 5. Related Works

## 5.1. Large Language Model Pretraining

Modern large language models are typically pretrained as autoregressive next token predictors on web-scale corpora, and scaling up in parameters, data and compute, leading to strong general purpose models (Brown et al., 2020; Kaplan et al., 2020; Hoffmann et al., 2022). Within this autoregressive setting, prior work has explored architectural and

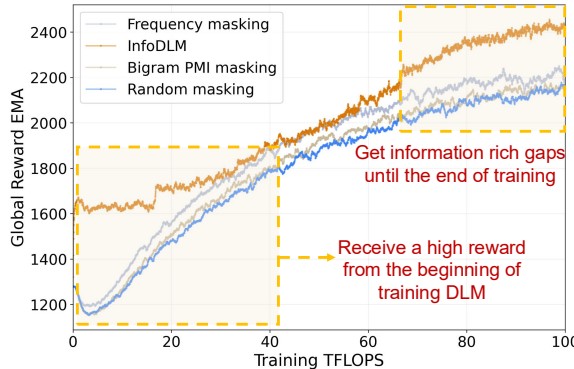

*Figure 4.* TIG-Reward under different policies when training DLM.

systems-level (Shazeer et al., 2017; Lewis et al., 2020b; Dai et al., 2019). More alternative sequence objectives arise that go beyond left to right prediction through infilling, span denoising and unified multitask mixtures (Fedus et al., 2022; Borgeaud et al., 2022; Lewis et al., 2020a; Raffel et al., 2020; Tay et al., 2022). More recently, several studies treat pretraining itself as a reinforcement learning problem on unlabeled corpora, and show that such reinforcement style pretraining can strengthen reasoning without extra labels (Dong et al., 2025; Li et al., 2025; Hatamizadeh et al., 2025). Together, these advances provide a mature and continually evolving backbone for autoregressive pretraining.

## 5.2. Diffusion Large Language Models

Diffusion language models (DLMs) depart from autoregressive next token prediction by coupling pretraining to a stochastic masking process that reconstruct text from partial observations. Diffusion-based language modeling includes continuous diffusion in an embedding space and discrete, masking-based diffusion directly over tokens for text generation.(Sohl-Dickstein et al., 2015; Ho et al., 2020; Song et al., 2021; Li et al., 2022; Austin et al., 2021). Recent discrete DLMs show that their training objectives can often be written as weighted masked language modeling losses (Li et al., 2022; He et al., 2023; Austin et al., 2021; Lou et al., 2024; Shi et al., 2024; Sahoo et al., 2024). This view has enabled masked diffusion systems scale to billions of parameters with competitive performance (Nie et al., 2025b; Ye et al., 2025; Nie et al., 2025a; Prabhudesai et al., 2025). To further improve DLM pretraining, techniques from autoregressive pretraining offer limited leverage and are not directly applicable, as they are tied to left-to-right sequential factorization. We instead focus on mask construction in this work, a new design axis apart from AR-LLM.

## 5.3. Masking Strategies

Masking is a core mechanism in masked language modeling and diffusion-based generation. In discrete token settings, prior work largely falls into three categories: (1) heuristic or schedule-based masking that selects tokens uniformly at random (Devlin et al., 2019; Liu et al., 2019); (2) linguistic and structure aware masking over whole words, spans or entity level units (Joshi et al., 2020; Cui et al., 2021; Sun et al., 2020); (3) difficulty or information aware masking that uses statistics such as pointwise mutual information, concept-level curricula, or model accuracy to decide masked positions and masking budget (Levine et al., 2021; Sadeq et al., 2022; Kosmopoulou et al., 2025; Yang et al., 2023; Sanyal et al., 2023; Lee et al., 2022; Edman & Fraser, 2025; Zhang et al., 2025). Information-aware objectives such as information gain (Lindley, 1956; MacKay, 1992) have been used for data selection in LLM fine-tuning such as FisherSFT (Deb et al., 2025). However, comparable information-aware principles have not been systematically applied to DLM pretraining, where masking typically remains static and ignores the model's evolving maturity. Consequently, existing DLM pretraining does not fully leverage masking to construct adaptive learning environments that deliver sustained capability gains. In this work, we treat masking as a central design knob of DLMs and jointly learn an information-rich, model-aware masking policy with the DLM.

## 5.4. Limitation and Future Work

Several limitations point to clear directions for future work. First, pretraining remains largely opaque, and our feedback signals provide only a partial view of how reasoning improve, calling for stronger diagnostics that link training dynamics to downstream generalization. Second, our evaluation is limited by compute and scale, and larger models with longer training budgets are needed to test whether the same mechanisms hold as a scalable behavior. Third, our theoretical understanding remains limited, and we need sharper conditions under which information-gain surrogates reliably track reasoning gains, along with large-scale validation.

Looking ahead, a natural step is to move from generic information gain to a reasoning-regret objective that targets reductions in reasoning-process error and can be coupled to downstream tasks. It will also be important to explore richer masking policies at the span or clause level and more stable RL variants that can exploit denser feedback with low overhead. Finally, extending information-gain–driven masking beyond diffusion-style text pretraining to autoregressive language models, code models, and multimodal or tool-augmented systems remains an open direction.

## 6. Conclusion

This work introduces INFODLM, a feedback-driven pretraining framework that optimizes the pretraining dynamics. At its core, the proposed TIG provides a single-step reward for updating the Mask Policy Network (MPN) via policy-gradient methods and is further aggregated into a maturity-aware signal for conditioning the masking policy. Empirically, this coupling offers a practical way to steer pretraining toward more informative corrupted samples while remaining lightweight, motivating further study of feedback-driven curriculum design for language model training.

## Acknowledgements

This work is supported by the U.S. Department of Energy, Office of Science, Office of Advanced Scientific Computing Research, for the DeCoDe project, in support of the MEER-CAT Microelectronics Science Research Center, under Contract DE-AC05-76RL01830. This work is also supported by DARPA under Contract W912CG25CA007, by NSF under Award No.2610649 and No.2326494, and by NERSC through DDR-ERCAP0035256.

## Impact Statement

This paper presents work whose goal is to advance the field of Machine Learning. There are many potential societal consequences of our work, none which we feel must be specifically highlighted here.

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

# SUMMARY OF THE APPENDIX

This appendix contains additional details for our submission. The appendix is organized as follows:

- §A provides pseudo code for the overall interleaved training procedure and the phase-conditioned mask policy.

- §B reports training details, including model hyperparameters, masking-policy training settings, dataset and evaluation protocols, and compute configurations.

- §C presents analyze of computation overhead introduced by Mask Policy Network.

- §F provides additional derivations and proofs for the TIG-based reward, including the last-layer gradient-energy identity and its masked-denoising instantiation.

- §D provides additional statistics of spaCy UPOS analsis, along with a quantitative analysis of context- and stage-adaptivity of the learned masking policy.

- §E provides ablations on the RL optimizer choice and the training-schedule sensitivity of the interleaved training scheme.

- §G formalizes the reasoning regret objective and gives the corresponding definition and proofs used in our analysis.

## A. Pseudo Code of InfoDLM and Code Release

The pseudo-code of InfoDLM is given in Algorithm 1. To guarantee reproducibility, our full implementation shall be publicly released upon paper acceptance.

## B. Training Details

**Data and preprocessing.**    We pretrain on SlimPajama with a sequence length of 2048. We apply standard tokenizer-based preprocessing and form training sequences by contiguous packing. For supervised fine-tuning, we apply the benchmarks in the following table.

**Model configuration.**    We use two model scales, summarized in Table 7.

*Table 7.* Model configurations used in pretraining.

| Model | #Layers | #Heads | $d_{\text{model}}$ | $d_{\text{ff}}$ |
|-------|---------|--------|---------|--------|
| 472M  | 18      | 10     | 1280    | 5120   |
| 1028M | 20      | 14     | 1792    | 7168   |

**Optimization and training schedule.**    We optimize both the DLM and the masking policy with AdamW under the interleaved schedule described in the main text. Unless otherwise stated, all reported *steps* and *tokens* refer to *DLM update* steps/tokens (policy-update steps are in addition and follow the same interleaving schedule). At 472M scale, we run 60k DLM steps on L40S GPUs with sequence length 2048, using total batch sizes of 280 for DLM blocks and 288 for policy blocks, corresponding to ∼35B DLM training tokens. At 1B scale, we run 204k DLM steps with total batch sizes of 576 for DLM blocks and 512 for policy blocks, corresponding to ∼205B DLM training tokens. Due to compute constraints, ablations are conducted only at 472M scale.

For the interleaving schedule, we first warm-start the mask policy network (MPN) and then alternate between DLM and policy blocks for a fixed number of cycles. At 472M scale, we train the MPN for 10k steps, followed by 10 cycles of

*Table 8.* pretraining schedules and optimization hyperparameters for all models. "Cycles" denotes the number of alternating DLM/MPN blocks after the initial MPN warm-up.

| Model | Params | Warm-up (MPN steps) | Cycles (DLM / MPN) steps per cycle | Total DLM steps | Total MPN steps | Tokens (B) | Global batch × seq length |
|---|---|---|---|---|---|---|---|
| INFODLM-1B | 1B | 10k | $22 \times (10k/2k)$ | 220k | 54k | $\approx 220$ | $576 \times 2048$ |
| Ablation-472M (A) | 472M | 8k | $12 \times (5k/1k)$ | 60k | 20k | $\approx 35$ | $288 \times 2048$ |
| Ablation-472M (B) | 472M | 8k | $12 \times (5k/1k)$ | 60k | 20k | $\approx 35$ | $288 \times 2048$ |
| INFODLM-472M | 472M | 8k | $12 \times (5k/1k)$ | 60k | 20k | $\approx 35$ | $288 \times 2048$ |
| Random-472M | 472M | 8k | $12 \times (5k/1k)$ | 60k | 20k | $\approx 35$ | $288 \times 2048$ |
| Span-472M | 472M | 8k | $12 \times (5k/1k)$ | 60k | 20k | $\approx 35$ | $288 \times 2048$ |
| Frequency-472M | 472M | 8k | $12 \times (5k/1k)$ | 60k | 20k | $\approx 35$ | $288 \times 2048$ |
| PMI-472M | 472M | 8k | $12 \times (5k/1k)$ | 60k | 20k | $\approx 35$ | $288 \times 2048$ |

alternating 5k DLM steps and 1k MPN steps. At 1B scale, we train the MPN for 15k steps, followed by 20 cycles of alternating 12k DLM steps and 2k MPN steps. We report the overall compute budget as an estimated total number of FLOPs. Detailed information can be found in Table 8.

**Evaluation protocol.**    For supervised fine-tuning, we use a total batch size of 896 with sequence length 256. We fine-tune for 2 epochs on ANLI, 2 epochs on SNLI. On GSM8K, for 1B model we train 40 epochs, and for 472M model, we train for 30 epochs.

## C. Computation Overhead Analysis

We estimate the FLOP overhead of INFODLM. Let the corpus contain $N$ tokens. Denote by $F_{\mathrm{fwd}}$ the per-token FLOPs of a DLM forward pass, by $F_{\mathrm{bwd}}$ the FLOPs of a full backward pass, by $F_{\mathrm{llwd}}$ the FLOPs of the last-layer backward pass, by $F_{\mathrm{MPN}}$ the FLOPs of one MPN forward+backward step, by $F_{\mathrm{ft}}$ the FLOPs of calculating the extra token feature. At pretraining, we use 20% of corpus to train the MPN, which is $0.2N$ tokens. Empirically we observe the rough ratios

$$F_{\mathrm{fwd}} \simeq 0.5 F_{\mathrm{bwd}}, \qquad F_{\mathrm{llwd}} \simeq 0.03 F_{\mathrm{bwd}}, \qquad F_{\mathrm{MPN}} \simeq 0.01 F_{\mathrm{bwd}}, \qquad F_{\mathrm{ft}} \simeq 0.05 F_{\mathrm{bwd}}.$$

As a reference, a diffusion LM with random masking performs one forward and one backward pass per token. The total cost over $N$ tokens is therefore

$$\mathrm{FLOPS}_{\mathrm{random}} = N\big(F_{\mathrm{fwd}} + F_{\mathrm{bwd}}\big) \simeq N \cdot 1.5 F_{\mathrm{bwd}}.$$

We now consider the computation overhead of INFODLM. When updating the MPN, the per-token cost consists of one DLM forward pass, an extra token feature calculation, a last-layer backward pass to obtain the gradient proxy, and an MPN forward+backward step:

$$\text{MPN update (per token)}: \qquad F_{\mathrm{fwd}} + F_{\mathrm{ft}} + F_{\mathrm{llwd}} + F_{\mathrm{MPN}}.$$

When updating the DLM under a fixed MPN, we pay for one DLM forward, token feature calculation, one full backward, and an MPN forward pass,

$$\text{DLM update (per token)}: \qquad F_{\mathrm{fwd}} + F_{\mathrm{ft}} + F_{\mathrm{bwd}} + F_{\mathrm{MPN\text{-}fwd}},$$

where $F_{\mathrm{MPN\text{-}fwd}}$ denotes the FLOPs of the MPN forward pass only. Suppose the MPN is updated on a fraction $0.2N$ of the tokens, while the DLM is updated on all $N$ tokens. The total cost over $N$ tokens is then

$$
\begin{aligned}
\mathrm{Total}_{\mathrm{last\text{-}layer}} &= 0.2N\big(F_{\mathrm{fwd}} + F_{\mathrm{ft}} + F_{\mathrm{llwd}} + F_{\mathrm{MPN}}\big) + N\big(F_{\mathrm{fwd}} + F_{\mathrm{ft}} + F_{\mathrm{bwd}} + F_{\mathrm{MPN\text{-}fwd}}\big) \\
&= N\big(1.2 F_{\mathrm{fwd}} + 1.2 F_{\mathrm{ft}} + 0.2 F_{\mathrm{llwd}} + F_{\mathrm{bwd}} + 0.53 F_{\mathrm{MPN}}\big) \\
&\simeq N\big(1.662 F_{\mathrm{bwd}} + 0.53 F_{\mathrm{MPN}}\big) \\
&\simeq N \cdot 1.6673 F_{\mathrm{bwd}} \\
&\simeq 1.11 \cdot \mathrm{FLOPS}_{\mathrm{random}}
\end{aligned}
$$

where in the last line we substitute the empirical ratios above and approximate $F_{\text{MPN-fwd}}$ as a fixed fraction of $F_{\text{MPN}}$.

Across masking strategies, INFODLM is the second-fastest in training throughput, with the speed ranking: random $>$ span $>$ INFODLM $>$ frequency $>$ PMI.

**Measured per-step cost.**    We also provide directly measured per-step costs at the exact 472M pretraining setting ($4\times$A100, DLM batch $24\times2$, MPN batch $64\times1$, phase-switch overhead $<2$ ms). In addition to the breakdown in Table 1 of the main text, Table 9 compares against an AR-472M autoregressive baseline and a full-gradient variant of TIG that replaces the last-layer proxy with a complete backward pass. The full-gradient ablation costs an additional $\sim36\%$ over our last-layer proxy without delivering a corresponding accuracy gain (see Sec. 4), supporting our design choice. The per-cycle aggregation (last column) follows the 5:1 DLM/MPN schedule used throughout.

*Table 9.* Measured per-step compute at 472M, including an autoregressive baseline (AR-472M) and a full-gradient variant of TIG. INFODLM (TIG, ours) uses the last-layer proxy; INFODLM (Full-Grad) materializes the full-model gradient instead.

| Method | Phase | Time/step (s) | Peak mem (GB) | FLOPs/step ($\times10^{14}$) | Cycle time (s) |
|---|---|---|---|---|---|
| AR-472M | — | 1.88 | 71.7 | 9.31 | 375.63 |
| SMDM (baseline) | DLM | 2.02 | 63.2 | 9.32 | 403.56 |
| INFODLM (TIG, ours) | DLM | 2.09 (+3.5%) | 68.6 (+8.6%) | 9.99 | 417.96 |
| INFODLM (TIG, ours) | MPN | 1.07 | 57.1 | 0.186 | 42.76 |
| INFODLM (Full-Grad) | DLM | 2.27 (+12.4%) | 71.3 (+12.8%) | 13.9 | 454.05 |
| INFODLM (Full-Grad) | MPN | 2.35 (+119.6%) | 64.0 | 0.408 | 94.06 |

Under the 5:1 DLM/MPN schedule, INFODLM incurs only $\sim14.1\%$ wall-clock overhead per cycle, whereas the full-gradient TIG variant adds a further $\sim36\%$ cost on top. AR training is $\sim7.4\%$ more compute-efficient per step than SMDM at this scale, which sets the reference against which our matched-compute comparison in the experiments section is calibrated.

## D. Additional Analysis on spaCy UPOS

**POS-aware masking behavior.**    To better understand what the learned mask policy chooses to hide, we analyze mask probabilities by part-of-speech (POS). We tag each token with spaCy's Universal POS (UPOS) tags (De Marneffe et al., 2021), which provide a coarse-grained syntactic category for every token: adjectives (ADJ), adpositions or prepositions (ADP), adverbs (ADV), auxiliary verbs (AUX), coordinating conjunctions (CCONJ), determiners (DET), interjections (INTJ), common nouns (NOUN), numerals (NUM), particles (PART), pronouns (PRON), proper nouns (PROPN), punctuation marks (PUNCT), subordinating conjunctions (SCONJ), symbols (SYM), main verbs (VERB), miscellaneous tokens (X), and whitespace tokens (SPACE). We then group these tags into five coarse semantic categories: (i) *reasoning-related* tokens (VERB, AUX, ADV, SCONJ); (ii) *entity/content* words (NOUN, PROPN, ADJ); (iii) *function words* (ADP, DET, PRON); (iv) *punctuation/layout* (PUNCT, SPACE); and (v) *miscellaneous symbols* (all remaining tags, including NUM, PART, CCONJ, INTJ, SYM, X).

At the end of training, InfoDLM-472M assigns higher per-token mask probabilities to reasoning and content-bearing categories than to layout tokens. Aggregated over the evaluation corpus, reasoning-related tags such as adverbs, verbs, adjectives and subordinating conjunctions (e.g., "if", "while") receive the highest mask scores (e.g., ADV $\approx 0.786$, ADJ/SCONJ/VERB $\approx 0.73$), followed by pronouns and common nouns. In contrast, punctuation and whitespace are masked much less often (PUNCT $\approx 0.32$, SPACE $\approx 0.29$), with other miscellaneous tags (symbols, numerals, particles, unknown tokens) in between. This pattern indicates that the policy prefers to erase tokens that carry semantic or logical content, rather than low-information formatting tokens.

To study how this preference evolves during training, we track the fraction of the mask probability mass allocated to each of the five coarse categories at several checkpoints for InfoDLM-472M. Starting from 4k MPN steps, the policy already assigns roughly 17% of its mask budget to reasoning-related tokens and 46% to entity/content words, with function words, punctuation/layout, and miscellaneous symbols accounting for the remaining 37%. As training proceeds, the share on reasoning-related tokens steadily increases from 16.97% (4k) to 21.42% (101.3k), while punctuation/layout decreases from about 10.4% to 7.8% and function words remain relatively stable around 16–17%. The mass on entity/content words fluctuates in a narrow band around mid-40% (e.g., 45.7% at 4k, 46.8% at 8k, 45.6% at 101.3k), and miscellaneous symbols stay close to 9–10% throughout. Overall, the MPN gradually shifts mask probability away from punctuation and formatting

*Table 10.* Quantitative masking dynamics across training stages (cycle 1, cycle 7, cycle 11). Jaccard overlap measures token-set similarity with static baselines. Context variance measures how much the same word's mask probability varies across different contexts. Inter-stage variance measures how much it varies across training stages.

|  | Rand | Freq | PMI | c1 | c7 | c11 |
|---|---|---|---|---|---|---|
| Jaccard vs Freq | 0.333 | 1.00 | — | 0.517 | 0.509 | 0.570 |
| Jaccard vs PMI | 0.333 | — | 1.00 | 0.332 | 0.324 | 0.309 |
| Context var. | 0.000 | 0.000 | — | 0.083 | 0.087 | **0.106** |
| Inter-stage var. | 0.000 | 0.000 | — | — | — | **0.032** |

toward verbs, auxiliaries, adverbs and content words, suggesting that the co-evolving policy learns to focus its denoising budget on tokens that are most diagnostic for the underlying reasoning structure of the sequence.

**Quantitative masking behavior.** The UPOS analysis above reveals *what* token types the MPN prioritizes. We next ask *how* these preferences arise: are they static corpus-level statistics, or do they depend on context and training stage? Table 10 reports three metrics across training cycles for the MPN, compared against the static baselines from Table 3.

Three observations emerge. First, the MPN's overlap with frequency-based masking increases during training ($0.517 \rightarrow 0.570$), indicating that it captures corpus-level token importance; yet $43\%$ of its masking decisions diverge from static frequency statistics, suggesting additional context-dependent criteria. In contrast, overlap with PMI masking slightly decreases ($0.332 \rightarrow 0.309$), suggesting that MPN learns a model-dependent notion of informativeness rather than corpus-level co-occurrence statistics. Second, context variance grows steadily across cycles and reaches $0.106$ by cycle 11, meaning the same word receives substantially different mask probabilities depending on its surrounding context, a property that no static strategy possesses. Third, only the MPN exhibits nonzero inter-stage variance ($0.032$), confirming that its masking strategy evolves as the DLM matures. Together, these results show that MPN operates along two dynamic axes, context-adaptive and stage-adaptive, that static baselines fundamentally cannot match, providing quantitative grounding for the phase conditioning mechanism in Sec. 3.2.

# E. RL Optimizer and Training Schedule Ablations

**RL optimizer ablation.** To validate our choice of REINFORCE over more complex policy-gradient methods, we compare three policy optimization variants under the same 3-cycle compute budget in Table 11. REINFORCE with an EMA running baseline achieves the best performance on both GSM8K and SNLI, while PPO offers no clear advantage. As shown in Figure **??**, removing the EMA baseline leads to substantially larger policy-loss fluctuations and higher variance, confirming that the EMA baseline is the main stabilizing component.

*Table 11.* RL optimizer ablation (472M, 3 cycles). Best in **bold**.

| Method | GSM8K | SNLI |
|---|---|---|
| InfoDLM-REINFORCE-EMA | **48.67** | **71.38** |
| InfoDLM-PPO | 47.54 | 66.31 |
| InfoDLM-REINFORCE-NoBaseline | 48.22 | 66.13 |

**Training schedule sensitivity.** To verify robustness to the interleaved training schedule, we compare four configurations under the same total DLM budget (15k steps) in Table 12. Performance varies by less than 0.9 points on GSM8K and 2.3 points on SNLI across all schedules, indicating low sensitivity. Shorter cycles yield smoother dynamics, while longer MPN blocks produce larger transient jumps at cycle boundaries.

# F. Proof of TIG Reward Proxy

**Definition of Information Gain.** Information gain is first formalized by Lindley (**?**) as the expected Kullback–Leibler (KL) divergence from prior to posterior, given an experimental design $e$. The information provided by a single observation $x$ from experiment $e$ is the KL increase of the posterior over the prior, averaged over the marginal $p(x \mid e)$:

$$\mathbb{E}_{x \sim p(x|e)}[\mathrm{KL}(p(\phi \mid x, e) \,\|\, p(\phi))].$$

*Table 12.* Training schedule sensitivity (472M, 15k DLM steps). Best in **bold**.

| Config | Warm | DLM | MPN | Cycle | GSM8K | SNLI |
|---|---|---|---|---|---|---|
| Default | 10k | 5k | 1k | 3 | **48.67** | **71.38** |
| Short cycles | 5k | 2.5k | 0.5k | 6 | 48.14 | 69.70 |
| Longer MPN | 10k | 5k | 2k | 3 | 48.82 | 69.93 |
| No cycle | 20k | 15k | 0 | 0 | 47.92 | 69.17 |

This quantity is exactly the mutual information between the parameter $\phi$ and the data $X$ under design $e$:

$$I(\phi; X \mid e) = \mathbb{E}_{x \sim p(x|e)}[\mathrm{KL}(p(\phi \mid x, e) \,\|\, p(\phi))] = \mathrm{KL}(p(\phi, x \mid e) \,\|\, p(\phi)\, p(x \mid e)).$$

MacKay (**?**) brought Lindley's definition into neural networks and active learning and explicitly termed it *information gain*. For a dataset $D$ (single or multiple observations), the same principle can be written as

$$I(\phi; D) = \mathbb{E}_D[\mathrm{KL}(p(\phi \mid D) \,\|\, p(\phi))].$$

**Information Gain.** From a Bayesian perspective, observing a dataset $D$ updates a prior $p(\phi)$ to a posterior $p(\phi \mid D) \propto p(D \mid \phi)\, p(\phi)$, and the information contributed by $D$ about the parameters is measured by the information gain

$$\mathrm{IG}(D) = \mathrm{KL}\big(p(\phi \mid D) \,\|\, p(\phi)\big). \tag{22}$$

Viewing $D$ as random, the expected reduction in parameter uncertainty is the mutual information between $\phi$ and $D$,

$$I(\phi; D) = \mathbb{E}_D\big[\mathrm{IG}(D)\big] = \mathbb{E}_D\left[\mathrm{KL}\big(p(\phi \mid D) \,\|\, p(\phi)\big)\right], \tag{23}$$

which provides an information-theoretic target that can be related to the Fisher geometry under appropriate local approximations.

**Fisher–Gaussian KL Connection.** We work with a conditional model $p_\phi(x_{\mathcal{M}} \mid x_{\backslash \mathcal{M}})$ that predict origin text $x_{\mathcal{M}}$ given masked text $x_{\backslash \mathcal{M}}$. Under a local Laplace approximation around a reference point $\phi_0$, and with a Gaussian prior $p(\phi) = \mathcal{N}(\phi_0, \Sigma_0)$, the posterior given a dataset $D = \{x^{(n)}\}_{n=1}^N$ is approximated by a Gaussian whose covariance is

$$\Sigma_{\mathrm{post}}(D) \approx \left(\Sigma_0^{-1} + \sum_{n=1}^N \mathcal{I}_{\mathrm{cond}}(\phi_0 \mid x_{\backslash \mathcal{M}}^{(n)})\right)^{-1}, \tag{24}$$

where the *conditional Fisher information* is defined by

$$\mathcal{I}_{\mathrm{cond}}(\phi \mid x_{\backslash \mathcal{M}}) = \mathbb{E}_{x_{\mathcal{M}} \sim p_\phi(\cdot | x_{\backslash \mathcal{M}})}\Big[\nabla_\phi \log p_\phi(x_{\mathcal{M}} \mid x_{\backslash \mathcal{M}}) \nabla_\phi \log p_\phi(x_{\mathcal{M}} \mid x_{\backslash \mathcal{M}})^\top\Big]. \tag{25}$$

Ignoring the mean shift between posterior and prior (which is second order under small updates around $\phi_0$), the information gain reads as the Gaussian KL with a common mean.

**Information Gain in Small Disturb by a Datum.** Assume that there is only a a slight drift of weight when training on a Datum that $\phi_0$ remains unchanged and slight difference in covariance , for symmetric positive-definite (SPD) matrices $\Sigma_0, \Sigma_1 \in \mathbb{R}^{d \times d}$,

$$\mathrm{KL}\big(\mathcal{N}(\phi_0, \Sigma_1) \,\|\, \mathcal{N}(\phi_0, \Sigma_0)\big) = \tfrac{1}{2}\left[\mathrm{Tr}(\Sigma_0^{-1}\Sigma_1) - d + \log \frac{\det \Sigma_0}{\det \Sigma_1}\right]. \tag{26}$$

Applying (26) with $\Sigma_1 = \Sigma_{\mathrm{post}}(D)$ and averaging over random datasets $D$ yields a mutual-information approximation whose dominant dependence on $\phi_0$ is through the log-determinant of the accumulated conditional Fisher,

$$I(\phi; D) \approx C + \tfrac{1}{2} \mathbb{E}_D\left[\log \det\left(\Sigma_0^{-1} + \sum_{n=1}^N \mathcal{I}_{\mathrm{cond}}(\phi_0 \mid x_{\backslash \mathcal{M}}^{(n)})\right)\right], \tag{27}$$

where $C$ is a constant independent of $\phi_0$. Thus, locally, maximizing mutual information amounts to increasing the eigenvalues of the conditional Fisher. To use a scalar surrogate that is easy to estimate, we relate $\log \det$ to the trace under a spectral-band assumption and PSD perturbations.

The detailed deduction is as folllow:

$$
\begin{aligned}
I(\phi; D) &\approx \mathbb{E}_D\Big[\mathrm{KL}\big(\mathcal{N}(\phi_0, \Sigma_{\mathrm{post}}(D)) \,\|\, \mathcal{N}(\phi_0, \Sigma_0)\big)\Big] \\
&= \frac{1}{2}\,\mathbb{E}_D\left[\mathrm{Tr}\left(\Sigma_0^{-1}\Sigma_{\mathrm{post}}(D)\right) - d + \log\frac{\det\Sigma_0}{\det\Sigma_{\mathrm{post}}(D)}\right] \\
&= \frac{1}{2}\,\mathbb{E}_D\left[\mathrm{Tr}\left(\Sigma_0^{-1}\Sigma_{\mathrm{post}}(D)\right) - d + \log\det\Sigma_0 - \log\det\Sigma_{\mathrm{post}}(D)\right] \\
&= \frac{1}{2}\,\mathbb{E}_D\left[\mathrm{Tr}\left(\Sigma_0^{-1}\Sigma_{\mathrm{post}}(D)\right) - d + \log\det\Sigma_0 + \log\det\left(\Sigma_{\mathrm{post}}(D)^{-1}\right)\right] \\
&\overset{(24)}{\approx} \frac{1}{2}\,\mathbb{E}_D\left[\mathrm{Tr}\left(\Sigma_0^{-1}\Sigma_{\mathrm{post}}(D)\right) - d + \log\det\Sigma_0 + \log\det\left(\Sigma_0^{-1} + \sum_{n=1}^{N}\mathcal{I}_{\mathrm{cond}}(\phi_0 \mid x_{\setminus\mathcal{M}}^{(n)})\right)\right] \\
&= C + \frac{1}{2}\,\mathbb{E}_D\left[\log\det\left(\Sigma_0^{-1} + \sum_{n=1}^{N}\mathcal{I}_{\mathrm{cond}}(\phi_0 \mid x_{\setminus\mathcal{M}}^{(n)})\right)\right],
\end{aligned}
\tag{28}
$$

where

$$
C := \frac{1}{2}\,\mathbb{E}_D\left[\mathrm{Tr}\left(\Sigma_0^{-1}\Sigma_{\mathrm{post}}(D)\right) - d + \log\det\Sigma_0\right]
\tag{29}
$$

is a constant that does not depend on the log-determinant of the accumulated conditional Fisher information and can therefore be absorbed in subsequent approximations.

**From Information Gain to Fisher Det.**

**Lemma F.1** (Spectral-band logdet–trace sandwich). *Let $A, B \in \mathbb{R}^{d \times d}$ be SPD with $B - A = \Delta \succeq 0$ and suppose there exist scalars $0 < \underline{\lambda} \leq \overline{\lambda}$ such that $\underline{\lambda}I \preceq A \preceq \overline{\lambda}I$ and $\underline{\lambda}I \preceq B \preceq \overline{\lambda}I$. Then*

$$
\frac{1}{\overline{\lambda}}\,\mathrm{Tr}(\Delta) \;\leq\; \log\det B - \log\det A \;\leq\; \frac{1}{\underline{\lambda}}\,\mathrm{Tr}(\Delta).
\tag{30}
$$

*Proof.* Consider $f(t) = \log\det(A + t\Delta)$ on $t \in [0, 1]$. By Jacobi's formula, $f'(t) = \mathrm{Tr}\big((A + t\Delta)^{-1}\Delta\big)$. The spectral-band assumption implies $\frac{1}{\overline{\lambda}}I \preceq (A + t\Delta)^{-1} \preceq \frac{1}{\underline{\lambda}}I$ for all $t \in [0, 1]$. Hence $\frac{1}{\overline{\lambda}}\mathrm{Tr}(\Delta) \leq f'(t) \leq \frac{1}{\underline{\lambda}}\mathrm{Tr}(\Delta)$. Integrating from 0 to 1 gives (30). $\qquad\square$

To connect the mutual-information surrogate in (27) with the trace of the Fisher information, we define

$$
A := \Sigma_0^{-1}, \quad \Delta(D) := \sum_{n=1}^{N}\mathcal{I}_{\mathrm{cond}}(\phi_0 \mid x_{\setminus\mathcal{M}}^{(n)}), \quad B(D) := A + \Delta(D),
$$

so that (27) can be rewritten as

$$
\begin{aligned}
I(\phi; D) &\approx C + \frac{1}{2}\, \mathbb{E}_D\big[\log\det\big(A + \Delta(D)\big)\big] \\
&= C + \frac{1}{2}\, \mathbb{E}_D\big[\log\det A \;+\; \big(\log\det\big(A + \Delta(D)\big) - \log\det A\big)\big] \\
&= C' + \frac{1}{2}\, \mathbb{E}_D[\log\det B(D) - \log\det A] \\
&\overset{\text{Lemma F.1}}{\geq} C' + \frac{1}{2}\, \mathbb{E}_D\left[\frac{1}{\overline{\lambda}}\, \mathrm{Tr}\big(\Delta(D)\big)\right] \\
&\overset{\text{Lemma F.1}}{\leq} C' + \frac{1}{2}\, \mathbb{E}_D\left[\frac{1}{\underline{\lambda}}\, \mathrm{Tr}\big(\Delta(D)\big)\right],
\end{aligned}
$$

where $C' := C + \frac{1}{2}\log\det A$ is a constant independent of the Fisher increments $\Delta(D)$ and

$$
\mathrm{Tr}\big(\Delta(D)\big) = \sum_{n=1}^{N} \mathrm{Tr}\,\mathcal{I}_{\mathrm{cond}}(\phi_0 \mid x_{\backslash\mathcal{M}}^{(n)}).
$$

Inequality (30) thus shows that, within a fixed spectral band and along PSD directions, increasing $\mathrm{Tr}\big(\Delta(D)\big)$ increases the log-determinant in (27) up to explicit multiplicative constants. Therefore, the trace offers a stable, computation-friendly surrogate objective that is locally aligned with mutual information.

Noting that $\mathrm{Tr}(\mathcal{I}) = \mathbb{E}\big[\|\nabla_\phi \log p_\phi(\cdot)\|_2^2\big]$, the conditional variant induced by (25) follows from the standard Fisher trace identity:

$$
\begin{aligned}
\mathrm{Tr}\,\mathcal{I}_{\mathrm{cond}}(\phi \mid x_{\backslash\mathcal{M}}) &= \mathrm{Tr}\, \mathbb{E}_{x_{\mathcal{M}} \sim p_\phi(\cdot \mid x_{\backslash\mathcal{M}})}\Big[\nabla_\phi \log p_\phi(x_{\mathcal{M}} \mid x_{\backslash\mathcal{M}}) \nabla_\phi \log p_\phi(x_{\mathcal{M}} \mid x_{\backslash\mathcal{M}})^\top\Big] \\
&= \mathbb{E}_{x_{\mathcal{M}} \sim p_\phi(\cdot \mid x_{\backslash\mathcal{M}})}\Big[\mathrm{Tr}\Big(\nabla_\phi \log p_\phi(x_{\mathcal{M}} \mid x_{\backslash\mathcal{M}}) \nabla_\phi \log p_\phi(x_{\mathcal{M}} \mid x_{\backslash\mathcal{M}})^\top\Big)\Big] \\
&= \mathbb{E}_{x_{\mathcal{M}} \sim p_\phi(\cdot \mid x_{\backslash\mathcal{M}})}\Big[\big\|\nabla_\phi \log p_\phi(x_{\mathcal{M}} \mid x_{\backslash\mathcal{M}})\big\|_2^2\Big],
\end{aligned}
$$

which yields

$$
\mathrm{Tr}\,\mathcal{I}_{\mathrm{cond}}(\phi \mid x_{\backslash\mathcal{M}}) = \mathbb{E}_{x_{\mathcal{M}} \sim p_\phi(\cdot \mid x_{\backslash\mathcal{M}})}\Big[\big\|\nabla_\phi \log p_\phi(x_{\mathcal{M}} \mid x_{\backslash\mathcal{M}})\big\|_2^2\Big]. \tag{31}
$$

A mask-selection policy $\pi_\theta(m \mid x)$ then naturally leads to the information-theoretic training objective

$$
\max_\theta\; \mathbb{E}_{x \sim D}\, \mathbb{E}_{m \sim \pi_\theta(\cdot \mid x)}\Big[\mathrm{Tr}\,\mathcal{I}_{\mathrm{cond}}(\phi \mid x_{\backslash\mathcal{M}})\Big], \tag{32}
$$

which, under the Laplace approximation, PSD aggregation, and spectral-band conditions used above, locally increases the mutual-information surrogate in (27).

**Design Trainable Information Gain Reward.** Start from the aforementioned conditional Fisher trace:

$$
\mathrm{Tr}\,\mathcal{I}_{\mathrm{cond}}(\phi \mid x_{\backslash\mathcal{M}}) = \mathbb{E}_{x_{\mathcal{M}} \sim p_\phi(\cdot \mid x_{\backslash\mathcal{M}})}\Big[\big\|\nabla_\phi \log p_\phi(x_{\mathcal{M}} \mid x_{\backslash\mathcal{M}})\big\|_2^2\Big]
$$

is the expectation, under the model's conditional distribution, of the squared $\ell_2$-norm of the score $\nabla_\phi \log p_\phi(x_{\mathcal{M}} \mid x_{\backslash\mathcal{M}})$. In a modern autodiff implementation, one would approximate this expectation by Monte Carlo: draw several samples $x_{\mathcal{M}}^{(k)} \sim p_\phi(\cdot \mid x_{\backslash\mathcal{M}})$, form the conditional log-likelihood

$$
\ell^{(k)}(x, m) \;:=\; \log p_\phi\big(x_{\mathcal{M}}^{(k)} \mid x_{\backslash\mathcal{M}}\big) = \sum_{i \in \mathcal{M}} \log p_\phi\big(x_i^{(k)} \mid x_{\backslash\mathcal{M}}\big),
$$

backpropagate $\ell^{(k)}$ through the entire network to obtain the full parameter gradient $g^{(k)} = \nabla_\phi \ell^{(k)}$, and then estimate

$$
\mathrm{Tr}\,\mathcal{I}_{\mathrm{cond}}(\phi \mid x_{\backslash\mathcal{M}}) \;\approx\; \frac{1}{K}\sum_{k=1}^{K} \big\|g^{(k)}\big\|_2^2.
$$

Even if we reduce to a single sample ($K = 1$), this procedure requires one full backward pass *per* $(x, m)$ pair and explicitly materializing the gradient vector over all parameters $\phi$, which is computationally infeasible when $\phi$ contains hundreds of millions of weights and the masking policy must be evaluated at scale.

**From conditional Fisher trace to an ideal per-sample reward.** A standard Monte Carlo reduction replaces the conditional expectation in Equation (31) by the realized masked content $x_{\mathcal{M}}^{\text{obs}}$ from the dataset, yielding the single-sample score-energy estimator

$$R_{\text{ideal}}(x, m) := \left\| \nabla_\phi \log p_\phi\left(x_{\mathcal{M}}^{\text{obs}} \mid x_{\backslash \mathcal{M}}\right) \right\|_2^2. \tag{33}$$

If $x_{\mathcal{M}}^{\text{obs}} \sim p_\phi(\cdot \mid x_{\backslash \mathcal{M}})$, then

$$\mathbb{E}\left[R_{\text{ideal}}(x, m) \mid x_{\backslash \mathcal{M}}\right] = \operatorname{Tr} \mathcal{I}_{\text{cond}}(\phi \mid x_{\backslash \mathcal{M}}),$$

so $R_{\text{ideal}}$ is an unbiased Monte Carlo estimator of the conditional Fisher trace. Under mild model–data mismatch, $R_{\text{ideal}}$ remains at most weakly biased, because $\mathbb{E}[\nabla_\phi \log p_\phi] = 0$ and $\operatorname{Tr} \mathcal{I} = \mathbb{E}[\|\nabla_\phi \log p_\phi\|_2^2]$. In a network, computing $R_{\text{ideal}}(x, m)$ amounts to: (i) forming the conditional log-likelihood of the ground-truth masked tokens,

$$\ell(x, m) := \log p_\phi\left(x_{\mathcal{M}}^{\text{obs}} \mid x_{\backslash \mathcal{M}}\right) = \sum_{i \in \mathcal{M}} \log p_\phi(x_i^{\text{obs}} \mid x_{\backslash \mathcal{M}}),$$

by a single forward pass; (ii) invoking autodiff to compute the full gradient $g(x, m) = \nabla_\phi \ell(x, m)$; and (iii) taking its squared norm $\|g(x, m)\|_2^2$ as the reward. This is already cheaper than estimating the full Fisher, but still scales linearly with the number of parameters and is therefore too expensive to use as a per-sample signal inside an outer reinforcement-learning loop.

**Last-layer gradient as a tractable proxy.** To obtain a cheaper surrogate, we exploit the structure of the final prediction layer. Consider the last affine layer with representation $h \in \mathbb{R}^d$, parameters $(W, b)$, logits $z = Wh + b$, probabilities $p = \operatorname{softmax}(z)$, and a one-hot target $e_y$ for token $y$. With cross-entropy loss $\ell = -\langle e_y, \log p \rangle = -\log p_y$, the gradient admits the closed form

$$\nabla_W \ell = (p - e_y)h^\top, \qquad \nabla_b \ell = p - e_y, \tag{34}$$

where $\nabla_W \ell \in \mathbb{R}^{V \times d}$ and $\nabla_b \ell \in \mathbb{R}^V$ for a vocabulary of size $V$. The Frobenius energy contributed by $W$ is then

$$\|\nabla_W \ell\|_F^2 = \left\|(p - e_y)h^\top\right\|_F^2 = \|p - e_y\|_2^2 \|h\|_2^2, \tag{35}$$

using the fact that the Frobenius norm of a rank-one matrix equals the product of the norms of its two factors. The joint last-layer energy adds the bias term $\|\nabla_b \ell\|_2^2 = \|p - e_y\|_2^2$, which can be absorbed into Equation (35) by appending a constant 1 to the representation, i.e., by replacing $h$ with $\tilde{h} = [h; 1]$ and $W$ with the corresponding augmented weight matrix.

Aggregating over the masked index set $\mathcal{M}$ in a sequence yields the computable last-layer proxy

$$R_{\text{proxy}}(x, m) = \sum_{i \in \mathcal{M}} \|p^{(i)} - e_{y_i}\|_2^2 \|h^{(i)}\|_2^2, \tag{36}$$

where $(p^{(i)}, h^{(i)}, y_i)$ are the prediction distribution, last-layer representation, and target token at position $i$, all obtained from a *single* forward pass of the frozen DLM. The term $\|p^{(i)} - e_{y_i}\|_2^2$ can be expanded as

$$\|p^{(i)} - e_{y_i}\|_2^2 = \sum_j (p_j^{(i)} - e_{y_i,j})^2 = 1 - 2p_{y_i}^{(i)} + \sum_j \left(p_j^{(i)}\right)^2 = 1 - 2p_{y_i}^{(i)} + \|p^{(i)}\|_2^2,$$

where $\sum_j p_j^{(i)} = 1$ and $e_{y_i, y_i} = 1$. The term $\|p^{(i)}\|_2^2 = \sum_j (p_j^{(i)})^2$ can be computed exactly or approximated by a top-$K$ partial sum over the largest probabilities; when the softmax tail is nearly flat, truncating to top-$K$ induces a mild downward bias but significantly reduces variance and computation.

**Why the proxy tracks the ideal Fisher-based reward.** The link between Equation (36) and Equation (33) follows from two standard facts. First, the Fisher trace identity states that

$$\operatorname{Tr} \mathcal{I}_{\text{cond}}(\phi \mid x_{\backslash \mathcal{M}}) = \mathbb{E}\left[\|\nabla_\phi \log p_\phi(x_{\mathcal{M}}^{\text{obs}} \mid x_{\backslash \mathcal{M}})\|_2^2 \mid x_{\backslash \mathcal{M}}\right] = \mathbb{E}\left[R_{\text{ideal}}(x, m) \mid x_{\backslash \mathcal{M}}\right].$$

Second, writing the parameters as $\phi = \{\phi_1, \ldots, \phi_L\}$ for different layers, the squared gradient decomposes as

$$\|\nabla_\phi \ell\|_2^2 = \sum_{\ell=1}^{L} \|\nabla_{\phi_\ell} \ell\|_2^2.$$

Moreover, by the chain rule, each layer gradient can be expressed as

$$\nabla_{\phi_\ell} \ell = J_\ell(x, m; \phi)^\top \nabla_z \ell,$$

where $J_\ell$ is the Jacobian of the logits $z$ with respect to $\phi_\ell$ and $\nabla_z \ell = p - e_y$ for cross-entropy. When we operate in a short-step fine-tuning regime around $\phi_0$, the upstream Jacobians $J_\ell$ for $\ell < L$ typically vary slowly across nearby datapoints and masks, while the prediction error $(p - e_y)$ encodes most of the curvature and variability. Under the resulting *last-layer dominance* assumption—namely that the across-sample variance of $\|\nabla_\phi \ell\|_2^2$ is primarily explained by the final readout layer—we have

$$R_{\text{ideal}}(x, m) = \|\nabla_\phi \ell(x, m)\|_2^2 \approx c(x, m) \|\nabla_{W,b} \ell(x, m)\|_2^2,$$

for some positive scaling factor $c(x, m)$ that depends weakly on $(x, m)$. The last-layer energy $\|\nabla_{W,b} \ell\|_2^2$ is precisely the quantity captured (up to an explicit reparameterization of $(W, b)$) by Equation (35) and its aggregation in Equation (36). Consequently, the random variable $R_{\text{proxy}}(x, m)$ in Equation (36) is positively correlated with $R_{\text{ideal}}(x, m)$ in Equation (33); in the limiting case where upstream Jacobians are exactly constant across $(x, m)$, the two are equivalent up to a global multiplicative constant, and ranking mask patterns by $R_{\text{proxy}}$ is identical to ranking them by the conditional Fisher-based reward $R_{\text{ideal}}$.

# G. Proof of Relationship between IG and Reasoning

**Definition of Reasoning.** The dictionary definition of "reasoning" is the action of thinking about something in a logical and sensible way. A reasoning model is a large language model (LLM) that has been fine-tuned to break complex problems into smaller steps, often called "reasoning traces," prior to generating a final output. It is most accurate to say that reasoning LLMs are trained to "show their work" by generating a sequence of tokens (words) that resembles a human thought process, and that this act of "verbalizing" thoughts seems to unlock latent reasoning capabilities that LLMs implicitly learn from their massive corpus of training data, which contains examples of individuals directly and indirectly articulating their own processes. These latent reasoning capabilities primarily come from pretraining.

**Strictly Proper Scoring Rule.** A scoring rule assigns a numerical score to a predictive distribution $P$ when an outcome $y$ is observed. Let the outcome space be finite $\mathcal{Y} = \{1, \ldots, V\}$ and let $P \in \Delta^{V-1}$ denote a predictive distribution. A scoring rule is a map

$$S: \Delta^{V-1} \times \mathcal{Y} \to \mathbb{R}, \qquad (P, y) \mapsto S(P, y),$$

with *expected score* under the *true data distribution* $Q \in \Delta^{V-1}$ given by

$$S(P, Q) := \mathbb{E}_{Y \sim Q} [S(P, Y)].$$

We say $S$ is **proper** if $S(Q, Q) \geq S(P, Q)$ for all $P \in \Delta^{V-1}$, i.e., reporting the true $Q$ maximizes the expected score; and **strictly proper** if equality holds only when $P = Q$. This guarantees *truthful* probabilistic predictions and, in particular, promotes calibration.

**Brier Score, one of the Strictly Proper Scores.** For discrete outcomes, write $e_y \in \{0, 1\}^V$ for the one-hot vector of $y$. The *Brier loss* (lower is better) and its corresponding scoring rule (higher is better) are

$$L_{\text{Brier}}(P, y) := \|P - e_y\|_2^2, \qquad S_{\text{Brier}}(P, y) := -\|P - e_y\|_2^2.$$

Under Brier Score, we usually use the follow two equations:

$$\|P - e_y\|_2^2 = 1 - 2P_y + \|P\|_2^2 \quad \text{and} \quad \mathbb{E}_{Y \sim Q} [\|P - e_Y\|_2^2] = \|P - Q\|_2^2 + \mathbb{E}_{Y \sim Q} [\|Q - e_Y\|_2^2].$$

And from the second equation, we could deduce why it's **Strict** and **Proper**: for any $Q$,

$$S_{\text{Brier}}(Q, Q) - S_{\text{Brier}}(P, Q) = \mathbb{E}_{Y \sim Q} [\|P - e_Y\|_2^2 - \|Q - e_Y\|_2^2] = \|P - Q\|_2^2 \geq 0,$$

with equality iff $P = Q$. We will use this strictly proper score in the following content.

**Definition of *Regret*.** In decision theory, *regret* (excess risk, Bayes regret) is the increase in expected loss of a reported decision relative to the Bayes-optimal decision. We extend this notion for *reasoning* by taking the verifiable target $\omega$ to be the object of reasoning (e.g., $\omega = y$ for the final answer, or $\omega = (y, \pi)$ for answer+process). Let $p_{\phi^\star}(\cdot \mid x, e)$ be the true conditional distribution and $q_{D,e}(\omega \mid x) = \int p_\phi(\omega \mid x, e)\, p(\phi \mid D, e)\, d\phi$ be the posterior predictive under design $e$. For a strictly proper scoring rule $S$ (higher is better), the *reasoning regret* is

$$\boxed{\mathcal{R}_S(D, e) = \mathbb{E}_x\Big[ \underbrace{\mathbb{E}_{\omega \sim p_{\phi^\star}}[-S(q_{D,e}, \omega)]}_{\text{model's expected loss}} - \underbrace{\mathbb{E}_{\omega \sim p_{\phi^\star}}[-S(p_{\phi^\star}, \omega)]}_{\text{Bayes-optimal baseline}} \Big] \geq 0.}$$

This is the classical decision-theoretic regret specialized to reasoning targets. Two canonical instantiations are: (i) log-score, yielding $\mathcal{R}_S = \mathbb{E}_x \mathrm{KL}\big(p_{\phi^\star} \| q_{D,e}\big)$; and (ii) Brier (squared) score, yielding $\mathcal{R}_S = \mathbb{E}_x \|q_{D,e} - p_{\phi^\star}\|_2^2$. Smaller $\mathcal{R}_S$ indicates better reasoning on the specified target and admits second-order bounds in terms of the posterior covariance in the LAN/BvM regime.

### G.1. Redefine Reasoning under *Reasoning Regret*

We formalize *reasoning ability* as reasoning regret over verifiable targets, conditioned on experiment design $e$ (e.g., masking policy, augmentation, sampling strategy, etc.). Let the ground-truth distribution be $p_{\phi^\star}$ and the posterior predictive distribution be

$$q_{D,e}(y, \pi \mid x) = \int p_\phi(y, \pi \mid x, e)\, p(\phi \mid D, e)\, d\phi,$$

where $x$ is the input, $y$ the final answer, and $\pi$ an optional process/program (derivation steps, proof traces, executable code, denotes "Chain-of-thought"). There are three types of reasoning:

1. **Outcome-only reasoning.** Using a strictly proper scoring rule $S_y$ on $y$ (e.g., log or Brier), we define the reasoning regret

$$\mathcal{R}_{S_y}(D, e) = \mathbb{E}_x\Big[\mathbb{E}_{y \sim p_{\phi^\star}}[-S_y(q_{D,e}(\cdot \mid x), y)] - \mathbb{E}_{y \sim p_{\phi^\star}}[-S_y(p_{\phi^\star}(\cdot \mid x, e), y)]\Big] \geq 0.$$

   For log-score, $\mathcal{R}_{S_y} = \mathbb{E}_x \mathrm{KL}(p_{\phi^\star} \| q_{D,e})$; for Brier, $\mathcal{R}_{S_y} = \mathbb{E}_x \|q_{D,e} - p_{\phi^\star}\|_2^2$.

2. **Process-aware reasoning.** When both answer and process are verifiable, we use a joint strictly proper score $S_{y,\pi} = S_y(P_Y, y) + \lambda\, S_{\text{step}}(P_{\Pi|Y}, \pi \mid y)$ with $\lambda > 0$, and define

$$\mathcal{R}_{S_{y,\pi}}(D, e) = \mathbb{E}_x\Big[\mathbb{E}_{(y,\pi) \sim p_{\phi^\star}}[-S_{y,\pi}(q_{D,e}, (y, \pi))] - \mathbb{E}_{(y,\pi) \sim p_{\phi^\star}}[-S_{y,\pi}(p_{\phi^\star}, (y, \pi))]\Big] \geq 0.$$

   Non-differentiable verifiers $V(x, \pi, y)$ can be upper-bounded by a convex surrogate $\psi \geq \mathbf{1}\{V = 0\}$ absorbed into $S_{\text{step}}$.

3. **Functional reasoning.** For tasks that *compute* a target functional $g$ (of parameters or predictive distributions), let $\widehat{g}_{D,e}$ be the estimator (e.g., posterior mean or an executable evaluation). We measure

$$\mathrm{MSE}_g(D, e) = \mathbb{E}_x \mathbb{E}\big[\|\widehat{g}_{D,e}(x) - g_{\phi^\star}(x)\|^2\big].$$

   Under standard regularity, these three risks provide verifiable and optimizable objectives that will admit second-order bounds in terms of the posterior covariance, enabling the subsequent link from information gain to reasoning improvement.

### G.2. How to Link Them Together?

There are connections between Reasoning Regret and the Posterior Covariance that we mentioned in Information inspired relationship in training a model to fit the data.

**Lemma G.1** (Second-order expansion of reasoning regret)**.** *Fix a design $e$ and an input $x$. Let*

$$\mathcal{L}_S(\phi; x, e) := \mathbb{E}_{\omega \sim p_{\phi^\star}(\cdot|x,e)}\big[-S\big(p_\phi(\cdot \mid x, e), \omega\big)\big],$$

*and assume: (i) the model is locally well-specified so that $\nabla_\phi \mathcal{L}_S(\phi^\star; x, e) = 0$; (ii) $\mathcal{L}_S(\cdot; x, e)$ is twice continuously differentiable in a neighborhood of $\phi^\star$ with Hessian $\mathcal{H}_S(x, e) := \nabla_\phi^2 \mathcal{L}_S(\phi^\star; x, e) \succeq 0$; (iii) a local asymptotic normality/BvM condition holds so that the posterior is concentrated and admits a second-order delta-method expansion: $\phi = \phi^\star + \Delta$, $\mathbb{E}[\Delta \mid D, e] = 0$, $\mathrm{Cov}[\Delta \mid D, e] = \Sigma_{\mathrm{post}}(D, e)$, and $\mathbb{E}[\|\Delta\|^3] = o(\|\Sigma_{\mathrm{post}}\|)$. Then*

$$\mathbb{E}_{\phi|D,e}\big[\mathcal{L}_S(\phi; x, e) - \mathcal{L}_S(\phi^\star; x, e)\big] = \tfrac{1}{2}\mathrm{tr}\big(\mathcal{H}_S(x, e)\,\Sigma_{\mathrm{post}}(D, e)\big) + o(\|\Sigma_{\mathrm{post}}\|). \tag{37}$$

*Consequently, the reasoning regret satisfies*

$$\boxed{\mathcal{R}_S(D, e) = \tfrac{1}{2}\,\mathbb{E}_x\,\mathrm{tr}\big(\mathcal{H}_S(x, e)\,\Sigma_{\mathrm{post}}(D, e)\big) + o(\|\Sigma_{\mathrm{post}}\|).} \tag{38}$$

*Proof.* By a second-order Taylor expansion of $\mathcal{L}_S(\cdot; x, e)$ at $\phi^\star$,

$$\mathcal{L}_S(\phi^\star + \Delta; x, e) = \mathcal{L}_S(\phi^\star; x, e) + \underbrace{\nabla_\phi \mathcal{L}_S(\phi^\star; x, e)^\top \Delta}_{=0} + \tfrac{1}{2}\Delta^\top \mathcal{H}_S(x, e)\,\Delta + r(\Delta),$$

where the remainder $r(\Delta) = o(\|\Delta\|^2)$ by smoothness. Taking the posterior expectation and using $\mathbb{E}[\Delta \mid D, e] = 0$ and $\mathbb{E}[\Delta\Delta^\top \mid D, e] = \Sigma_{\mathrm{post}}(D, e)$ yields

$$\mathbb{E}_{\phi|D,e}\big[\mathcal{L}_S(\phi; x, e) - \mathcal{L}_S(\phi^\star; x, e)\big] = \tfrac{1}{2}\mathrm{tr}\big(\mathcal{H}_S(x, e)\,\Sigma_{\mathrm{post}}(D, e)\big) + \mathbb{E}[r(\Delta)].$$

The delta-method condition implies $\mathbb{E}[r(\Delta)] = o(\|\Sigma_{\mathrm{post}}\|)$. Finally, take the outer expectation over $x$ and recall that

$$\mathcal{R}_S(D, e) = \mathbb{E}_x\Big(\mathbb{E}_{\phi|D,e}\big[\mathcal{L}_S(\phi; x, e)\big] - \mathcal{L}_S(\phi^\star; x, e)\Big).$$

$\square$

Let $\Sigma_0 \succ 0$ be the prior covariance and $F_e(D) \succeq 0$ the (data-aggregated) Fisher information under design $e$ and dataset $D$. Define the Laplace posterior covariance

$$\Sigma_{\mathrm{post}}(D, e) := \big(\Sigma_0^{-1} + F_e(D)\big)^{-1}.$$

Let $S$ be a strictly proper scoring rule with local curvature $\mathcal{H}_S(x, e) \succeq 0$ at input $x$ (log-score: conditional Fisher). Define the quadratic upper bound of reasoning regret

$$U(D, e) := \tfrac{1}{2}\,\mathbb{E}_x\,\mathrm{tr}\big(\mathcal{H}_S(x, e)\,\Sigma_{\mathrm{post}}(D, e)\big).$$

Define the (Gaussian/Laplace) information gain

$$\mathrm{IG}(e) := \tfrac{1}{2}\,\mathbb{E}_D\,\log\det\Big(I + \Sigma_0^{1/2} F_e(D)\Sigma_0^{1/2}\Big).$$

**Lemma G.2** (Covariance order under Fisher order). *If $F_2(D) \succeq F_1(D)$ (Löwner order) for all $D$, then*

$$\Sigma_{\mathrm{post}}^{(2)}(D) = \big(\Sigma_0^{-1} + F_2(D)\big)^{-1} \preceq \big(\Sigma_0^{-1} + F_1(D)\big)^{-1} = \Sigma_{\mathrm{post}}^{(1)}(D).$$

*Hence,* more Fisher information $\Rightarrow$ smaller posterior covariance *(in Löwner order).*

**Lemma G.3** (Quadratic bound decreases with covariance). *For any positive semidefinite weight $\mathcal{H}_S(x, e) \succeq 0$,*

$$\Sigma_{\mathrm{post}}^{(2)} \preceq \Sigma_{\mathrm{post}}^{(1)} \quad\Longrightarrow\quad U^{(2)}(D, e) \leq U^{(1)}(D, e).$$

*Thus,* smaller posterior covariance $\Rightarrow$ smaller quadratic upper bound on reasoning regret.

**Lemma G.4** (Compatibility of IG with Fisher order). *If $F_2(D) \succeq F_1(D)$ for all $D$, then*

$$\mathrm{IG}_2 \geq \mathrm{IG}_1.$$

*Thus,* Fisher increase (Löwner) $\Rightarrow$ IG does not decrease.

**Corollary G.5** (Strong monotone chain). *If a design update satisfies $F_{\mathrm{new}}(D) \succeq F_{\mathrm{old}}(D)$ for all D, then, simultaneously:*

$$\mathrm{IG}_{\mathrm{new}} \ \geq \ \mathrm{IG}_{\mathrm{old}}, \qquad \Sigma_{\mathrm{post}}^{\mathrm{new}} \ \preceq \ \Sigma_{\mathrm{post}}^{\mathrm{old}}, \qquad U^{\mathrm{new}} \ \leq \ U^{\mathrm{old}}.$$

*Hence,* more information $\Rightarrow$ smaller posterior covariance $\Rightarrow$ smaller regret upper bound.

**Lemma G.6** (IG increase implies bound decrease under spectral alignment). *Let $X := \Sigma_0^{1/2} F_e(D) \Sigma_0^{1/2}$ and $B := \mathbb{E}_x\big[\Sigma_0^{1/2} \mathcal{H}_S(x,e) \Sigma_0^{1/2}\big]$. Assume X and B are (approximately) simultaneously diagonalizable with eigenvalues $\{\lambda_i\}_{i=1}^d$ and $\{b_i\}_{i=1}^d$, where $\lambda_i \in [0, U_{\max}]$ and $b_i \in [b_{\min}, b_{\max}]$. Then*

$$U(D,e) = \tfrac{1}{2} \sum_{i=1}^{d} \frac{b_i}{1 + \lambda_i}, \qquad \mathrm{IG}(e) = \tfrac{1}{2} \mathbb{E}_D \sum_{i=1}^{d} \log(1 + \lambda_i).$$

*For any coordinate-wise nonnegative increment $\Delta\lambda_i \geq 0$, one has*

$$\Delta U \ \leq \ - \frac{b_{\min}}{1 + U_{\max}} \, \Delta\mathrm{IG}.$$

*Thus,* under spectral alignment and band limits, increasing IG forces a proportional decrease of the regret upper bound.

**Therefore,** Either (i) ensure $F_{\mathrm{new}} \succeq F_{\mathrm{old}}$ (additive/convex design updates), which yields the *strong chain* IG $\uparrow \Rightarrow$ $\Sigma_{\mathrm{post}} \downarrow \Rightarrow U \downarrow$ by Lemmas G.2–G.4 and Cor. G.5; or (ii) enforce spectral alignment/regularization so that Lemma G.6 applies, giving a quantitative guarantee that *IG increase decreases the regret upper bound*.

**Algorithm 1** Pseudo-code of InfoDLM in a PyTorch-like style.

```
# DLM: p(x | z); Policy: pi(mask | x, t, phase)
# Global Metrics GM maintains: EMA entropy, Accumulative IG, update per 5 steps.

def TrainDLMStep(x, dlm, policy, gm):
    B,T,t = x.shape, Uniform(0,1,size=B)

    with no_grad():
        g_H,g_R = gm.get_phase(B, x.device) # phase snapshot
        embedding, feature = dlm.encode(x) # embeddings + shallow hidden
        p_raw, logits = policy(embedding, feature, t, g_H, g_R)
        p_mask_pol, _ = ProjectToRatio(logits, t) # enforce mask ratio ~ t; lambda
            discarded here (DLM update does not need it)

    z = ApplyMask(x, mask, p_mask_pol) # get bernoulli sampled mask based on p_mask_pol

    dlm_logits, dlm_hidden = dlm(z, return_hidden=True)
    L_ce = CrossEntropy(dlm_logits[mask], x[mask]) # only masked positions

    gR_step = TIGProxy(dlm_logits, dlm_hidden, x, mask) # step reward from same forward
    gH_step = Entropy(Softmax(dlm_logits[mask]))

    # (3) update DLM by CE, accumulate maturity stats (does NOT publish snapshot g)
    Optimize(dlm, L_ce)
    did_flush = gm.update_from_batch(mean(gH_batch), mean(gR_step)) # refresh running
        stats every K=5 steps
    return L_ce, mean(r_step)

def TrainMPNStep(x, dlm, policy, gm, baseline):
    B,T,t = x.shape, Uniform(0,1,size=B)
    g_H,g_R = gm.get_phase(B, x.device)

    embedding, feature = dlm.encode(x)
    p_raw, logits = policy(embedding, feature, t, g_H, g_R)
    p_mask_pol, lam = ProjectToRatio(logits, t) # lam: non-learnable per-sample bias
        from bisection s.t. sigmoid(logits+lam).mean() = t
    z = ApplyMask(x, mask, p_mask_pol)

    with no_grad():
        dlm_logits, dlm_hidden = dlm(z, return_hidden=True)
        r_step = TIGProxy(dlm_logits, dlm_hidden, x, mask)

    # Adding lam to logits inside the policy loss keeps the action distribution ratio-
        consistent with sampling.
    loss = PolicyLoss(r_step - baseline.update(mean(r_step)), logits+lam, mask, t) #
        Policy Gradient Loss
    Optimize(policy, loss)
    return loss

# ---------------- Main training (single interleaved schedule) ----------------
def Train(data_dlm, data_mpn, dlm, policy, mode, cycles, N_dlm, N_mpn, GM):
    if mode in ["mpn_only","alternating"]: WarmUpGlobalMetrics(data_dlm, dlm, GM,
        W_steps) # get initial g_H, g_R

    for c in range(cycles):
        for _ in range(N_dlm):
            x = next(data_dlm)
            TrainDLMStep(x, dlm, policy, GM) # gm freezes on every K-step flush
        gm.freeze_phase() # publish accumulated stats as snapshot g; held constant for
            the full MPN block
        for _ in range(N_mpn):
            x = next(data_mpn)
            TrainMPNStep(x, dlm, policy, GM, baseline)
```

