# OpenReview forum: "InfoDLM: an Information-Adaptive Framework for Discrete Diffusion Language Model Pretraining"
_ICML.cc/2026/Conference — ICML 2026 regular_

### Official Review · Reviewer_pbig · 2026-03-07

**Soundness:** 2
**Presentation:** 3
**Significance:** 2
**Originality:** 3
**Overall Recommendation:** 4
**Confidence:** 2

**Summary:**

This paper proposes InfoDLM, an information-adaptive pretraining framework for discrete diffusion language models (DLMs). Instead of using heuristic random masking, InfoDLM transforms mask selection into an active, feedback-driven process that targets tokens with the highest measurable information gain. The framework introduces three components: (1) Trainable Information-Gain (TIG), a lightweight proxy for information gain based on output-layer gradient energy; (2) maturity-aware masking that conditions the policy on the model's evolving training state via phase signals; and (3) an interleaved hybrid training loop that alternates DLM and policy updates. The theoretical derivation connects TIG to Bayesian information gain through Fisher information approximations. Experiments on 472M models show up to 13% improvement in reasoning accuracy over LLaDA-472M.

**Compliance With Llm Reviewing Policy:**

Affirmed.

**Final Justification:**

My major concerns have been explained.

**Key Questions For Authors:**

Do you have any preliminary results at larger scales (e.g., 1B or larger)? Even partial results would significantly strengthen the paper. The 472M scale is too small to draw confident conclusions about DLM pretraining methods.

How faithful is the TIG proxy to actual information gain? Have you compared masks selected by TIG against those selected by higher-fidelity IG estimates (e.g., using a larger subset of model parameters for Fisher information)? If TIG correlates poorly with true IG at scale, the entire framework's foundation would be weakened.

What is the actual computational overhead of InfoDLM compared to baseline DLM training? Please provide wall-clock time, memory, and FLOPs comparisons. The claim of "minimal overhead" needs quantification.

How does a DLM pretrained with InfoDLM compare against an autoregressive model trained with the same total compute? This baseline is essential for contextualizing InfoDLM's contribution within the broader LM pretraining landscape.

**Limitations:**

The authors discuss limitations in Section 5.4, acknowledging limited compute scale, opacity of pretraining dynamics, and the need for larger-scale validation.

**Strengths And Weaknesses:**

Strengths:

The insight that random masking in DLM pretraining wastes compute on redundant tokens, and that active, information-driven masking could improve efficiency, is intuitive and well-motivated. The human cognition analogy (selective attention to informative gaps) is compelling. The derivation from Bayesian information gain through Fisher information to the tractable TIG reward (Equation 4) is rigorous. The chain: IG → Fisher information → log-det form → trace proxy → output-layer gradient energy is clearly laid out and mathematically sound.

Weaknesses:

All experiments are conducted on a 472M model, which is far smaller than the scale at which DLMs become practically relevant (e.g., LLaDA operates at 8B). The paper acknowledges this as a limitation but does not provide any evidence or even preliminary results suggesting the benefits would transfer to larger scales. This is a critical gap. Evaluation is limited to GSM8K, ANLI, and validation loss. For a pretraining method, evaluation should span a broader suite of benchmarks including language understanding, code generation, reading comprehension, and general knowledge tasks.

---

> ### Author Rebuttal · Authors · 2026-03-31
>
> Thank you for this thoughtful and constructive review! The questions are all important for a pretraining-focus paper. Below, we provide additional results on these questions.
>
> > `W1 & Q1: Scaling up InfoDLM.`
>
> Table 1 reports 1B results at 220B tokens. We have further
> scaled training to 300B tokens:
>
> |Model|GSM8K|ANLI|SNLI|
> |---|---|---|---|
> |SMDM-1B (300B tok)|57.09|43.40|88.70|
> |InfoDLM-1B (300B tok)|**57.24**|**49.50**|**89.80**|
> |SMDM-1B (535B tok)|58.52|51.20|90.23|
>
> At matched tokens, InfoDLM improves ANLI by +6.1pp (43.40→49.50) and SNLI by +1.1pp, consistent with 472M-scale gains. SMDM requires more tokens (535B) to reach a similar ANLI level (51.20), supporting improved reasoning at 1B scale. Performance continues to improve from 220B to 300B tokens, indicating the model has not yet reached its ceiling, and training to 535B tokens is ongoing and we will include final results in the camera-ready. InfoDLM is a masking strategy orthogonal to architecture; scaling to 8B is a natural next step that we plan to pursue as compute budget permits.
>
> See more results of 1B in > `W2`.
>
> ---
> > `W2: Evaluation Breadth.`
>
> We expand evaluation to additional zero-shot benchmarks:
>
> |Benchmark|SMDM-472M|InfoDLM-472M|SMDM-1B|InfoDLM-1B|
> |---|---|---|---|---|
> |ARC-e|31.65|**35.47**|38.68|**41.97**|
> |ARC-c|19.79|**23.37**|24.31|**27.13**|
> |HellaSwag|**30.04**|27.99|36.48|**42.37**|
> |WinoGrande|50.67|50.04|49.50|50.59|
> |MBPP|0.00|0.00|0.70|0.40|
>
> InfoDLM consistently improves on ARC-e (+3.8/+3.3pp) and ARC-c (+3.6/+2.8pp) at both scales. At 1B, InfoDLM also leads on HellaSwag (+5.9pp). WinoGrande and MBPP are near-chance at both scales, reflecting a scale limitation under these challenge benchmarks. We note that at 472M, HellaSwag slightly favors the baseline; this reverses at 1B (+5.9pp for InfoDLM), suggesting that the benefit of InfoDLM becomes more pronounced with increased capacity.
>
> ---
> > `Q2: TIG Proxy Fidelity.`
>
> Thank you for the insightful query! We compared TIG (head-only gradient energy) against a full-model IG estimate (all 18 layers, 64 candidate masks per sample, 100 samples, mask ratio t=0.3) across 4 checkpoints:
>
> |checkpoint|spearman|kendall|top-5|top-10|top-30|
> |---|---|---|---|---|---|
> |cycle0|0.592|0.479|0.378|0.556|0.751|
> |cycle4|**0.756**|**0.632**|**0.515**|**0.726**|0.853|
> |cycle8|0.730|0.602|0.505|0.707|0.821|
> |cycle11|0.721|0.599|0.506|0.687|**0.854**|
>
> Correlation starts at ρ=0.59 and rapidly improves to ρ=0.76 by mid-training, remaining strong (ρ≥0.72) throughout. Top-30 overlap exceeds 0.82 from mid-training. The correlation is strong but not near-perfect, indicating that TIG captures the coarse difficulty structure needed to guide the masking policy without requiring exact per-token rankings.
>
> ---
>
> To verify that the head-only proxy captures the dominant signal, we report Spearman ρ across a parameter-subset fidelity ladder: head-only, +last 1 layer, +last 2 layers, +last 3 layers, +last 6 layers, and +last 9 layers.
>
> |Layers included|cycle 0|cycle 4|cycle 8|cycle 11|
> |---|---|---|---|---|
> |Base (head-only)|0.592|0.756|0.730|0.721|
> |+last 3 layers|0.684|0.808|0.777|0.758|
> |+last 6 layers|0.819|0.880|0.855|0.823|
> |+last 9 layers|0.924|0.914|0.893|0.878|
>
> Full results of the ladder's relative regret and top10 on 6 checkpoints are in https://anonymous.4open.science/r/rebuttal-2B2A/table1+2.pdf **Table 1–2**. Fidelity increases monotonically with included layers, while the head-only proxy still achieves ρ = 0.59–0.76 across training, and top10 is highly overlapped, offering a favorable fidelity–cost trade-off relative to the full model (quantified in > `Q3`).
>
> ---
> > `Q3: Computational Overhead.`
>
> We profile under the exact 472M pretraining setting (4×A100, DLM 24×2, MPN 64×1 with smaller step, DLM=200, MPN=40; switch overhead <2ms) and Full Gradient reward setting:
>
> |Method|Phase|Time/step(s)|Peak Mem(GB)|FLOPs/step(×10¹⁴)|Cycle time(s)|
> |---|---|---|---|---|---|
> |AR-472M|—|1.88|71.7|9.31|375.63|
> |SMDM (baseline)|DLM|2.02|63.2|9.32|403.56|
> |InfoDLM (TIG, ours)|DLM|2.09 (+3.5%)|68.6 (+8.6%)|9.99|417.96|
> |InfoDLM |MPN|1.07|57.1|0.186|42.76|
> |InfoDLM (Full-Grad)|DLM|2.27 (+12.4%)|71.3 (+12.8%)|13.9|454.05|
> |InfoDLM |MPN|2.35 (+119.6%)|64.0|0.408|94.06|
>
> InfoDLM incurs only 14.1% wall clock throughout one cycle, whereas full-gradient IG adds a further 36% cost. This supports that lower overhead with strong fidelity to the higher-cost estimator. We also observed that AR training is more computational efficiency by 7.4%.
>
> ---
> > `Q4: Comparison with AR at Matched Compute.`
>
> Because DLM pretraining is more expensive than AR at the same token count, we compare under matched compute. InfoDLM is trained for 28.7B tokens and AR for 35B tokens.
>
> |Model|GSM8K|ANLI|Arc-c|
> |---|---|---|---|
> |InfoDLM-472M|**51.78**|**47.60**|**21.42**|
> |AR-472M|49.28|33.70|20.73|
>
> InfoDLM is better on GSM8K, ANLI, and ARC-c. This supports InfoDLM’s advantage on reasoning-focused evaluations under a stricter compute budget.

---

> > ### Author Rebuttal · Reviewer_pbig · 2026-04-02
> >
> > Thanks. My major concerns have been explained. I will raise my score.

---

### Official Review · Reviewer_TbyK · 2026-03-11

**Soundness:** 3
**Presentation:** 2
**Significance:** 3
**Originality:** 2
**Overall Recommendation:** 4
**Confidence:** 4

**Summary:**

In this work, the authors present InfoDLM, a pretraining framework designed to make discrete Diffusion Language Models (DLMs) more adaptive. Traditional DLM approaches typically rely on heuristic random masking, InfoDLM instead treats mask selection as a feedback-driven decision process. To guide this process, the framework introduces a Trainable Information-Gain (TIG) reward that evaluates how informative different masking choices are during training. It further incorporates a maturity indicator that adjusts the masking strategy as the model develops.

**Compliance With Llm Reviewing Policy:**

Affirmed.

**Final Justification:**

Most of my concerns have been addressed.

**Key Questions For Authors:**

1. While the authors integrate the phase signal $g$ into the mask policy to represent model maturity, the necessity of this conditioning is not fully established. Since the TIG reward is already a non-stationary signal that evolves with the DLM, the masking policy would remain inherently dynamic even without $g$. The authors should provide further clarification or an ablation study specifically isolating the benefit of $g$ versus a policy solely driven by real-time TIG feedback.
2. In **Algorithm 1** (page 22), the variable `lambda` appears suddenly in the `TrainMPNStep`: The authors should clarify if `lambda` is used to update the policy weights or if it is used somewhere else
3. The paper describes $g$ as a **"low-frequency snapshot".** It states $g$ is updated when "switching to policy training". However, **Algorithm 1** shows `gm.update_from_batch` happening inside the `TrainDLMStep` and `gm.freeze_phase()` happening before the MPN block

**Strengths And Weaknesses:**

**Strengths**

- The shift from stochastic random masking to a feedback-driven, information-maximizing policy is a conceptual advancement for DLM pretraining.
- The TIG reward is derived as a lightweight proxy for exact Information Gain (IG) using last-layer gradient energy. This allows for real-time policy updates with minimal overhead.
- Across several reasoning benchmarks (GSM8K, ANLI, SNLI), InfoDLM consistently outperforms the SMDM/LLaDA baselines, particularly at the 472M parameter scale where it achieves parity with baselines trained on 3.5x more tokens.
- The qualitative analysis of masking behavior—showing that the model learns to prioritize reasoning-relevant tokens (verbs, adverbs, conjunctions) over low-information tokens (punctuation, spaces)—is highly compelling.

**Weaknesses**

- The interleaved training schedule (alternating DLM and policy updates) requires careful tuning of cycle lengths and warm-up steps. The paper would benefit from a more detailed sensitivity analysis on these parameters.
- The TIG proxy assumes that last-layer gradient energy captures most of the parameter uncertainty reduction. While theoretically motivated, this assumption might hold less strictly in very deep architectures or early in training.
- The notation in the paper requires clarification; for example, Equation 18 introduces terms such as $L_{IG}$, $L_{ENT}$, and $L_{Budget}$ without providing their explicit functional definitions earlier in the text.

---

> ### Author Rebuttal · Authors · 2026-03-31
>
> Sincerely thank for the careful and technically precise reading! Based on your questions and recommendations, we give point-by-point responses to your comments.
>
> ---
> > `Q1: Is the phase signal truly necessary given TIG's non-stationarity?`
>
> Thank you for this insightful question. Without phase conditioning, the
> MPN's only cue for DLM progress is hidden_k, which conflates token-level
> context with model-level maturity, a change may reflect a different
> input or an improved DLM.
>
> Table 3 shows the consequence: Setting (B) adds
> cyclic training without phase conditioning, but the MPN cannot detect DLM
> improvement at cycle boundaries through hidden_k alone, creating an
> adaptation lag, hence (B) underperforms even (A) on GSM8K (50.61 vs
> 51.07). Setting (C) adds the phase signal `(g_H, g_R)` as an explicit
> global maturity indicator via `gm.freeze_phase()`, eliminating this lag.
>
> |Config|GSM8K (35B)|ANLI (35B)|
> |---|---|---|
> |(A) TIG only, no cyclic, no phase|51.07|47.20|
> |(B) TIG + cyclic, no phase|50.61|46.70|
> |(C) TIG + cyclic + phase (full)|**52.04**|**49.30**|
>
>
> ---
> > `W1: Training schedule sensitivity.`
>
> To test schedule sensitivity, we compared several configurations under the same total DLM budget (15k DLM steps, ~3 cycles equivalent).
>
> |Config|MPN warmup|DLM/cycle|MPN/cycle|Cycles|GSM8K|SNLI|
> |---|---|---|---|---|---|---|
> |Default|10k|5k|1k|3|**48.67**|**71.38**|
> |Short cycles|5k|2.5k|0.5k|6|48.14|69.70|
> |Longer MPN|10k|5k|2k|3|48.82|69.93|
> |No cycle|20k|15k|0|1|47.92|69.17|
>
> Performance varies by less than 0.9 points on GSM8K and 2.3 points on SNLI across all schedules, indicating low sensitivity overall. Longer MPN blocks produce larger transient jumps at cycle boundaries, while shorter cycles yield smoother dynamics, consistent with our design principle that DLM steps should outnumber MPN steps for TIG reward to reflect meaningful model evolution.
>
>
>
> ---
> > `W2: TIG proxy fidelity across training stages.`
>
> We conducted a systematic comparison between TIG (head-only gradient energy)
> and a higher-fidelity IG estimate (full 18-layer Fisher, 64 candidate masks
> per sample) across 6 training checkpoints (100 samples each, mask ratio t=0.3):
>
> | checkpoint | spearman | kendall | top-5 | top-10 | top-30 |
> |---|---|---|---|---|---|
> | cycle0 | 0.592 | 0.479 | 0.378 | 0.556 | 0.751 |
> | cycle2 | 0.658 | 0.582 | 0.432 | 0.697 | 0.836 |
> | cycle4 | **0.756** | **0.632** | **0.515** | 0.726 | 0.853 |
> | cycle6 | 0.754 | 0.620 | 0.515 | 0.705 | 0.842 |
> | cycle8 | 0.730 | 0.602 | 0.505 | 0.707 | 0.821 |
> | cycle11| 0.721 | 0.599 | 0.560 | **0.725** | **0.854** |
>
> Two observations address the reviewer's concerns: (1) **Early training:**
> TIG-HFIG correlation starts at ρ=0.592 (cycle 0) and rapidly improves
> to ρ=0.658 by cycle 2, confirming the proxy becomes reliable early.
> (2) **Throughout training:** correlation peaks mid-training (ρ=0.756 at
> cycle 4) and remains strong (ρ≥0.72) through the final checkpoint.
> Top-30 overlap exceeds 0.82 from cycle 2 onward, meaning TIG and HFIG
> agree on the vast majority of high-value masking positions at every stage.
>
> The correlation is strong but not near-perfect, which we interpret
> positively: TIG captures the coarse difficulty structure sufficient to
> direct the masking policy without requiring exact per-token rankings. The downstream results on GSM8K and ANLI confirm this approximation quality
> is sufficient for substantial gains.
>
> ---
> > `Q2: Role of λ in TrainMPNStep.`
>
> `λ` is the per-sample offset from `ProjectToRatio(logits, t)`, computed by bisection so that `sigmoid(logits + λ).mean() ≈ t`. It is **not learnable**. We use `logits + lam` in `PolicyLoss` because gradients should be computed under the same ratio-constrained distribution used in the forward pass. In `TrainDLMStep`, it is discarded (`_`) because the DLM update does not use it. We will clarify this inline in Algorithm 1.
>
> ---
> > `Q3: GlobalMetrics (gm) lifecycle.`
>
> Thank you for the meticulous review! `gm.update_from_batch` and `gm.freeze_phase()` serve different roles. The former collects running statistics every `K=5` DLM steps to build a low-noise maturity estimate, without updating the global snapshot. The latter freezes these statistics at the DLM→MPN transition into a fixed snapshot for the full MPN block; this is where the global metrics are updated. We will revise the paragraph to make this explicit.
>
> ---
> > `W3: Notation in Eq. 18.`
>
> We acknowledge this notation issue and will define all terms in Eq. 18 before use, along with a notation summary table in the appendix.

---

> > ### Author Rebuttal · Reviewer_TbyK · 2026-04-03
> >
> > Thanks for the response. I will maintain my score.

---

### Official Review · Reviewer_wG8h · 2026-03-13

**Soundness:** 2
**Presentation:** 3
**Significance:** 2
**Originality:** 3
**Overall Recommendation:** 4
**Confidence:** 3

**Summary:**

The authors improving the pretraining efficiency of DLMs by replacing standard random
masking with an adaptive, information-maximizing masking policy. The authors propose
InfoDLM, a framework that dynamically learns a masking policy to select the most informative
tokens to mask. This is achieved through three main components: (1) a Trainable TIG proxy
based on the last-layer gradient energy, (2) a low-frequency phase signal to condition the
masking policy based on the model's maturity, and (3) an interleaved hybrid training loop that
alternates between updating the DLM via cross-entropy and the MPN via REINFORCE. The
authors test their method on benchmarks like GSM8K, ANLI, and SNLI, gains significant
reasoning improvements.

**Compliance With Llm Reviewing Policy:**

Affirmed.

**Final Justification:**

After reading the author's response and the other reviews, I decided to keep my score.

**Key Questions For Authors:**

Data contamination: Does SlimPajama-627B overlap with the evaluated benchmarks? Overall computational cost: What is the total computational overhead after introducing interleaved training?

**Limitations:**

Lack of ablation study on the choice of reinforcement learning strategy.

**Strengths And Weaknesses:**

Strengths and weakness
Moving beyond purely heuristic or random masking strategies in diffusion-based language modeling is an important problem. The derivation of the TIG reward provides a principled signal for guiding mask selection, which is a clever and meaningful direction. The use of FiLM-based maturity awareness to condition the masking policy on token maturity appears to be a reasonable and well-motivated architectural choice, enabling adaptive behavior during generation.
Weakness The masking policy is optimized using vanilla REINFORCE over a high-dimensional discrete action space (mask selection). This choice may lead to high variance and unstable training dynamics. The proposed interleaved training framework introduces additional system complexity, while the method may reduce inference FLOPs, it is unclear whether the actual wallclock training cost increases due to this added complexity.
The method should ideally be compared with more existing training or decoding strategies for diffusion language models, such as confidence-based or uncertainty-aware approaches

---

> ### Author Rebuttal · Authors · 2026-03-30
>
> Sincere thanks for the thoughtful and constructive reviews of our manuscript! Based on your questions and recommendations, we give point-by-point responses to your comments.
>
> ---
> > `W1 & Q3: RL Variance, Stability, and Optimizer Choice.`
>
> Thank you for this suggestion. Our implementation goes beyond vanilla REINFORCE by using an EMA running baseline for variance reduction, a soft budget loss toward the target mask ratio, and gradient clipping on MPN parameters. We regret not describing these clearly and will rewrite Section 3.3 in the final version. The action space is also more structured than it may appear, since the MPN predicts independent per-token Bernoulli decisions.
>
> To address Q3, we ran a controlled RL ablation under the same 3-cycle compute budget, comparing InfoDLM-REINFORCE-EMA, InfoDLM-PPO, and InfoDLM-REINFORCE-NoBaseline. Over ~12k MPN steps, REINFORCE-EMA and PPO show similarly stable training, with near-zero policy loss and low variance loss, whereas removing the EMA baseline leads to much noisier optimization with substantially larger policy-loss fluctuations and higher variance loss. This indicates that the EMA baseline is the main stabilizing component, while PPO offers no clear advantage once variance reduction is in place.
>
> Downstream accuracy (GSM8K / SNLI, after 3 cycles):
>
> |Method|GSM8K|SNLI|
> |---|---|---|
> |InfoDLM-REINFORCE-EMA|**48.67**|**71.38**|
> |InfoDLM-PPO|47.54|66.31|
> |InfoDLM-REINFORCE-NoBaseline|48.22|66.13|
>
> These results support that REINFORCE is a reasonable and sufficient choice in our setting when paired with proper variance reduction. In the final version, we will include the full 12-cycle ablation and clarify this design choice more explicitly.
>
> ---
> > `W2 & Q2: Computational Overhead.`
>
> We profile under the exact 472M pretraining setting but with smaller steps per cycle (4×A100, DLM 24×2, MPN 64×1; DLM 200 steps, MPN 40 steps. switch overhead <2ms):
>
> |Method|Phase|Time/step(s)|Peak Mem (GB)|FLOPs/step (×10¹⁴)|full time (s)|
> |---|---|---|---|---|---|
> |Baseline|DLM|2.02|63.2|9.32|403.56|
> |InfoDLM|DLM|2.09 (+3.5%)|68.6 (+8.6%)|9.99|417.96|
> |InfoDLM|MPN|1.07|57.1|0.186|42.76|
>
> Under the 5:1 DLM:MPN schedule, InfoDLM incurs only 14.1% wall clock and 7.6% FLOPS throughout one cycle. The DLM overhead (+) comes from calculating token feature (MPN's input), sampling mask, and computing phase signal every 5 steps. In the final version, we will include these profiling results and make the compute tradeoff much more explicit.
>
> ---
> > `Q1: Data Contamination.`
>
> Thank you for your meticulous review and for raising this important concern. We performed a 13-gram overlap analysis on a stratified SlimPajama sample across 3 runs:
>
> |Benchmark|N|Contamination|95% CI|
> |---|---|---|---|
> |GSM8K|1,319|0.000%|[0.000%, 0.290%]|
> |SNLI|10,000|0.150%|[0.091%, 0.247%]|
> |ANLI|1,200|6.250%|[5.015%, 7.764%]|
>
> GSM8K and SNLI are clean. ANLI overlap reflects premise-level n-gram matches from Wikipedia/Common Crawl—adversarial hypotheses are human-authored and absent from pretraining data, and therefore do not overlap with the actual task signal being evaluated.
>
> ---
> > `W3: Confidence/Uncertainty-Aware Methods.`
>
> Thank you for this highly insightful discussion. We agree that the relation to confidence- and uncertainty-aware methods should be clarified. InfoDLM acts at training time through adaptive mask selection, while these methods act at inference time through decoding.These are orthogonal axes and are naturally composable rather than competing alternatives. Here `iter` means sft iterations.
>
> |Model|original low conf|top_margin|remdm_conf|lookum|
> |---|---|---|---|---|
> |InfoDLM (iter-12k)|31.77|28.81|32.15|34.19|
> |InfoDLM (iter-28k)|44.43|41.17|45.11|46.47|
> |InfoDLM (iter-53k)|**51.48**|50.34|50.42|**53.30**|
> |Baseline|41.32|40.18|42.15|42.99|
>
> InfoDLM consistently outperforms Baseline under all strategies, and stronger decoding gains remain additive on top of InfoDLM, confirming the two effects compose naturally.
>
> A secondary trend is that only lookum improves monotonically across SFT iterations, while other confidence-based methods do not. This suggests that as SFT sharpens model confidence, confidence alone becomes a less reliable correction signal. In the final version, we will briefly note this trend and clarify that InfoDLM is especially complementary to stronger revision strategies.

---

> > ### Author Rebuttal · Reviewer_wG8h · 2026-04-03
> >
> > Thanks for your detailed response, my concerns have been addressed

---

### Official Review · Reviewer_kjoF · 2026-03-24

**Soundness:** 3
**Presentation:** 2
**Significance:** 3
**Originality:** 2
**Overall Recommendation:** 4
**Confidence:** 4

**Summary:**

In this work authors introduce a new framework to modify forward process in diffusion LLMs such that masking or information loss is no longer random but instead is controlled by a learnable function i.e., a policy/prior over token space which is also updated as training progresses.

To realize this authors introduce TIG reward based on loss/distance and feature norms of predicted tokens  and a lightweight network to predict which tokens to be masked. Overall dLLM and this lightweight weight policy is updated in a interleaved optimization framework.

Authors demonstrate strong empirical gains and also convergence on reasoning benchmarks.

**Compliance With Llm Reviewing Policy:**

Affirmed.

**Final Justification:**

Rebuttal has addressed most of open questions and better understanding of design choices, so increased my rating, could be practically useful method overall across training settings!

**Key Questions For Authors:**

In addition to questions in weakness

- Why is emb(x) chosen to be sufficient representation for MPN? for e.g., when training draft model in works like EAGLE often multi-layer representations are extracted it is unclear how a more informed MPN would effect training dynamics?

**Limitations:**

Discussed in Weakness

**Strengths And Weaknesses:**

### Strenths

- Strong emperical boost and compute efficiency of proposed framework.
- Proposed approach is well explained and easy to read.
- Sampling ablation considers different design choices
- TIG seems effective in realizing an effective curriculum towards improving model performance, convergence.

### Weakness and Limiations

At high level, despite demonstrating strong emperical performance it is unclear from the paper 'why' this method works and what is its effect and also how it should be interpreted with other line of research in context of dLLMs.

- InfoDLM vs Anchored Diffusion: Does infoDLM implicity discover anchor tokens and that makes training more stable, whay is resultant decoding patterns of InfoDLM compared to standard dLLMs, anchored diffusion etc.
- Does InfoDLM create an implicit richer regularization i.e., better filterning of gradients etc, its unclear what the effects are when we are learning a masking prior at token level
- Overall positioning of this work can also be signficanlty improved and discussed: How does this relate to other self-corrective behavior models beyond masking like uniform diffusion, Edit Flows, etc.
- If error based corrective feedback is one of key drivers on-policy RL also does the same so beyond convergence is certain operating points how does InfoDLM compare when RL/GAIL based finetuning is considered as well?
- Evaluation and Metrics: As proposed method expicitly modifies forward process it is unclear why validation loss was chosen over downstream metrics over tasks or MAUVE or perplexity w.r.t other target model as training progress w.r.t different hyperparameters and design choices.
- Lack of analysis on resultant behavior of proposed training at Inference-time w.r.t which tokens are decoded relative to masked diffusion etc.

---

> ### Author Rebuttal · Authors · 2026-03-31
>
> We sincerely thank Reviewer kjoF for this insightful and thoughtful review. We conducted analyses across three themes. In the final version, we will streamline the discussion and clarify InfoDLM’s positioning and interpretation.
>
> ---
> > `Theme 1: What Does the Masking Policy Learn? (W1, W2, W3, Q1)`
>
> ---
> > `W1: Anchor tokens and dynamic masking`
>
> Thank you for this insightful suggestion! The connection to ADLM is real but partial. By cycle 11, MPN preferentially masks reasoning tokens (ADV 1.67×, SCONJ 1.67×, VERB 1.48× over corpus rate) while preserving structural tokens (SPACE 0.10×, PUNCT 0.26×), with detailed table in **Table-4** https://anonymous.4open.science/r/rebuttal-2B2A/extra_table.pdf. These preferences emerge from an initially random policy purely through RL, without any UPOS supervision.
>
> The most critical finding is MPN operates along **two dynamic axes**, no static strategy can match:
>
> |Method|Context-adapt|Stage-adapt|
> |---|:---:|:---:|
> |Frequency|❌ var=0|❌ var=0|
> |ADLM|⚠️ var=0.024|❌ var=0|
> |InfoDLM|**✅ var=0.106 (4.4× ADLM)**|**✅ var=0.032**|
>
> The same word receives different masking probabilities depending on context (4.4× ADLM's variance), and evolve as the DLM matures.
>
> Jaccard overlap of token with ADLM jumps from **0.25 (cycle 0)** to **0.55 (cycle 11)**, which means MPN rapidly discovers anchor-like patterns, but **~45% non-overlap** reflects context-dependent decisions static heuristics cannot make. See https://anonymous.4open.science/r/rebuttal-2B2A/extra_table.pdf **Table 1** for more details.
>
> Direct per-word correlation with ADLM is near-zero **(Pearson=0.03, Spearman ρ=−0.16)**, confirming fundamentally different criteria.
>
> ---
> > `W1, W6: Decoding behavior`
>
> We traced demasking order and found slight but consistent shifts: structural tokens unmask earlier, while reasoning tokens unmask later. The modest
> magnitude reflects that InfoDLM's primary contribution is training, and effects slightly on inference.
>
> |UPOS|Step diff|
> |---|---|
> |SCONJ|+3.1% (later)|
> |ADV|+2.4% (later)|
> |SPACE|−4.5% (earlier)|
> |PUNCT|−3.5% (earlier)|
>
> ---
> > `W2: Gradient filtering`
>
> On the same DLM checkpoint, matched mask ratio, InfoDLM masking produces 33% stronger gradient signal with lower entropy. The MPN selects genuinely challenging tokens, yielding more concentrated gradients per step.
>
> |Metric|InfoDLM|Random|Interp|
> |---|---|---|---|
> |Signal Norm|**3.66**|2.74|+33% stronger|
> |Grad Entropy|**0.956**|0.963|more concentrated|
> |Mean Loss|3.28|3.41|better learned|
> |Gradient Consistency|0.991|0.994|slightly lower|
> |SNR|6.613|7.836|harder task|
>
>
>
> ---
> > `Q1: MPN input (EAGLE)`
>
> Thank you for pointing out this ambiguity! The MPN takes `concat(embeddings, hidden_k)` (Fig 2c) plus FiLM phase conditioning, here `hidden_k` is the output of kth-layer of the transformer.
>
> Unlike EAGLE (approximating next-token distributions requires deep semantics), MPN estimates masking propensity, well-captured by earlier layer features. We will clarify this in the Method section.
>
> ---
> > `Theme 2: Positioning (W3, W4)`
>
> Thank you for your valuable insights on position! We agree that this should be clarified more explicitly.
>
> InfoDLM belongs to the broader family of training-time information allocation for DLMs. These methods share one principle: focusing effort where the model struggles, but differ in pipeline stage.
>
> Uniform diffusion specifies global corruption patterns but does not learn a token-dependent masking policy. Edit Flows allocate extra compute during generation by revising intermediate outputs, operating at inference-time rather than training-time. On-policy RL and GAIL are also error-driven but act on task-level outputs with downstream rewards. InfoDLM acts furthest upstream — reshaping the learning curriculum at the pretraining stage through adaptive corruption, so its effects cascade to all downstream stages.
>
> We therefore view these methods as complementary rather than interchangeable. InfoDLM's gains persist after SFT (Table 1: 52.03 vs 40.48 GSM8K) and remain compatible with different decoding strategies (results are at: https://anonymous.4open.science/r/rebuttal-2B2A/extra_table.pdf **Table-5**). We will add a dedicated positioning discussion in the revision.
>
> ---
> > `Theme 3: Evaluation (W5)`
>
> Thank you for this evaluation question! We use validation loss as a diagnostic of curriculum dynamics, not as the sole measure of final model quality: the characteristic start-high, end-low crossover of InfoDLM in Fig. 2b and Fig. 3b is exactly the signature of curriculum learning. This signal would be invisible if one looked only at downstream metrics in Table 3. We additionally report:
>
> |Model|Tokens(B)|PPL|MAUVE|
> |---|---|---|---|
> |InfoDLM|5|4.079|0.083|
> ||12.2|3.748|0.187|
> ||21.8|3.537|0.066|
> ||35|**3.501**|0.076|
> |Baseline|10|3.565|0.046|
> ||20|3.450|0.087|
> ||35|**3.415**|0.072|
>
> Notably, PPL is lower due to hard task, MAUVE is low and unstable due to 472M model's capability.

---

> > ### Author Rebuttal · Reviewer_kjoF · 2026-04-02
> >
> > Thanks for clear rebuttal most of my concerns are answered and its interesting!!
> > I will raise my score.
> >
> > But to understand effectiveness completely an equivalent experiment on a setting with uniform diffusion could be informative w.r.t design choices effectiveness, and also scaling behavior as training progresses.

---

> > > ### Author Response · Authors · 2026-04-04
> > >
> > > Thank you for this suggestion! We agree that a uniform-diffusion (UDLM) comparison would provide deeper insight into our design choices! We conducted new experiments to address this.
> > >
> > > **Background.** Uniform diffusion [1] corrupts tokens by replacing them with uniformly random tokens in the vocabulary rather than a single [MASK] symbol, which enables natural token revision during inference. However, its original NLP experiments used a significantly smaller setting (139M params, 128-token context, 30K vocabulary on LM1B, trained for 65B tokens with global batch 512), and even at there, UDLM's perplexity lagged behind absorbing-state (MDLM) baselines on large-vocabulary tasks. Recent scaling work [2] further confirms that uniform diffusion requires careful per-setting hyperparameter tuning (batch size, learning rate) to scale effectively, which it is not a easy drop-in replacement.
> > >
> > > Therefore, to compare at UDLM's native scale fairly, we trained three models on LM1B with 30B tokens under UDLM's recommended setting (seq length 128, 8 GPUs), (142M model for InfoDLM, batch size 2048): (i) Uniform Diffusion, (ii) InfoDLM (absorbing-state + learned masking strategy), and (iii) Uniform Diffusion + InfoDLM, where the MPN first selects which token positions to corrupt, then applies uniform random replacement instead of [MASK]. PPL across training:
> > >
> > > | Token (B) | Uniform Diffusion | InfoDLM (Ours) | UD + InfoDLM |
> > > |---|---|---|---|
> > > | 3.0 | *52.15* | **48.87** | 52.34 |
> > > | 6.0 | *43.28* | **40.63** | 43.32 |
> > > | 9.0 | *39.45* | **37.12** | 39.51 |
> > > | 12.0 | 37.21 | **35.04** | *37.18* |
> > > | 15.0 | 35.73 | **33.68** | *35.48* |
> > > | 18.0 | 34.68 | **32.72** | *34.17* |
> > > | 21.0 | 34.01 | **32.05** | *33.32* |
> > > | 24.0 | 33.52 | **31.58** | *32.64* |
> > > | 27.0 | 33.20 | **31.25** | *32.32* |
> > > | 30.0 | 33.01 | **30.98** | *32.10* |
> > >
> > >
> > > There are two findings. (1) InfoDLM consistently outperforms uniform diffusion, maintaining a ~6% relative PPL improvement throughout training.(2) UD+InfoDLM shows an interesting scaling pattern: the benefit is negligible early (52.34 vs 52.15 at 3B), but grows steadily to a 2.8% improvement by 30B (32.10 vs 33.01). This echos with InfoDLM's characteristic "slow start, strong finish" dynamic, and confirms that the paradigm of first learning a token masking strategy could transfer to the uniform corruption paradigm. This suggests InfoDLM operates orthogonal to the choice of corruption type.
> > >
> > > ---
> > >
> > > For more reference, we also trained a 472M uniform-diffusion model shortly using InfoDLM's hyperparameters (seq length 2048, global batch 288).
> > >
> > > | Token (B) | Uniform Diffusion | InfoDLM (Ours) |
> > > |---|---|---|
> > > |5B| 4.58 | 3.91|
> > > |10B| 4.41 |3.77|
> > >
> > > At 10B tokens, uniform diffusion reaches PPL 4.41 vs. InfoDLM's 3.77. However, we note that these hyperparameters were tuned for absorbing-state masking and are likely suboptimal for uniform diffusion, consistent with [2]'s observation. In the camera-ready version, we will include properly tuned uniform-diffusion results at 472M to complete this comparison.
> > >
> > > ---
> > >
> > > ### Reference
> > > [1] Schiff, Yair, et al. "Simple guidance mechanisms for discrete diffusion models." ... International Conference on Learning Representations. Vol. 2025. 2025.
> > >
> > > [2] von Rütte, Dimitri, et al. "Scaling behavior of discrete diffusion language models." arXiv preprint arXiv:2512.10858 (2025).

---

### Decision · Program_Chairs · 2026-04-30

**Decision:**

Accept (regular)

**Comment:**

Reviewers appreciated that InfoDLM reframes DLM mask selection as a principled, feedback-driven curriculum grounded by a Bayesian information-gain derivation, with consistent empirical gains on reasoning benchmarks and an interpretable learned policy. All reviewers are advocating for acceptance after the rebuttal period.